# Synthesis, Single Crystal X-ray Structure, Spectroscopy and Substitution Behavior of Niobium(V) Complexes Activated by Chloranilate as Bidentate Ligand

**Alebel Nibret Belay** [1,*]🆔, **Johan Andries Venter** [2,*], **Orbett Teboho Alexander** [2] and **Andreas Roodt** [2,*]🆔

1  Department of Chemistry, Bahir Dar University, Bahir Dar P.O. Box 79, Ethiopia
2  Department of Chemistry, University of the Free State, P.O. Box 339, Bloemfontein 9300, South Africa
*  Correspondence: alebel.nibret@bdu.edu.et (A.N.B.); venterja@ufs.ac.za (J.A.V.); aroodta@gmail.com (A.R.)

**Abstract:** Chloranilic acid (2,5-dichloro-3,6-dihydroxy-1,4-benzoquinone, caH$_2$) as a bidentate ligand for Nb(V) as a metal center is presented. The different coordination behavior of caH$_2$ is well illustrated by a monomeric (Et$_4$N)*cis*-[NbO(ca)$_2$(H$_2$O)OPPh$_3$]·3H$_2$O.THF (**5**) and a novel tetranuclear compound (Et$_4$N)$_4$[Nb$_4$O$_4$(ca)$_2$($\mu^2$-O)$_2$Cl$_8$]·2CH$_3$CN (**6**) via self-assembly, respectively. These were obtained in >80% yields and characterized by IR, UV/Vis and NMR ($^1$H, $^{13}$C{$^1$H}, $^{31}$P{$^1$H}) spectroscopy and single crystal X-ray diffraction, and they included a systematic assessment of the solid-state behavior. The anionic metal complexes showed different coordination modes at the Nb(V): [Nb$_4$O$_4$(ca)$_2$($\mu^2$-O)$_2$Cl$_8$]$^{4-}$ (**6a**; distorted octahedral) and *cis*-[NbO(ca)$_2$(H$_2$O)(OPPh$_3$)]$^-$ (**5a**; $D_{5h}$ distorted pentagonal bipyramidal), respectively. The tetranuclear complex **6a** is substitution inert, while *cis*-[NbO(ca)$_2$(H$_2$O)OPPh$_3$]$^-$ (**5a**) allowed a systematic ligation kinetic evaluation. The substitution of the coordinated triphenylphosphine oxide by a range of pyridine-type entering nucleophiles, 4-*N*,*N*-dimethyl-aminopyridine (DMAP), pyridine (py), 4-methylpyridine (4Mepy), 3-chloropyridine (3Clpy) and 3-bromopyridine (3Brpy) in acetonitrile at 31.2 °C was carefully evaluated. The subtle interplay between the main group ligand systems and the hard, early transition metal Nb(V) complex (**5a**) was well illustrated. The entering monodentate ligands showed a 15-fold reactivity range increase in the order 3Brpy < 3Clpy < 4Mepy < py < DMAP in broad agreement with the Brønsted-donating ability of the nucleophiles. The activation parameters determined for the reaction of **5a** with DMAP as the entering ligand yielded $\Delta H^{\neq}_{kf}$ = 52 ± 1 kJ mol$^{-1}$ and $\Delta S^{\neq}_{kf}$ = −108 ± 3 J K$^{-1}$ mol$^{-1}$ for the enthalpy and entropy of activation, respectively, indicating an associative substitution mechanism. The study presents an important contribution to the structure/reactivity relationships in Nb(V) complexes stabilized by chloranilic acid as a bidentate ligand.

**Keywords:** niobium; chloranilic acid; substitution; mechanism; kinetics; crystal structures

## 1. Introduction

Niobium and tantalum occur in nature almost entirely as single isotopes ($^{93}$Nb and $^{181}$Ta); they occur mostly in the pentavalent state and commonly substitute one another in minerals due to having similar ionic radii [1]. These similarities in chemical and physical properties are the reason for the difficult separation of the two elements during the processing of tantalum and niobium-containing minerals. These group 5 metals' behavior toward different ligands, which include mono-halido and pseudo halido as well as bidentate hard ligands with O,O'- and N,O-donor atoms were investigated in our group [2–5], but in the majority of cases, these compounds were synthesized under an inert atmosphere and favored a mono-chelated moiety [6–8]. The instability of these complexes under ambient conditions may be ascribed to the hydrolyzation of the metal synthon, e.g., NbCl$_5$ or TaCl$_5$ [9]. However, some exceptions are observed where mono-coordination of the

bidentate ligand had taken place in hydrous conditions, but this seems to be an exception rather than the norm [10]. Niobium and tantalum pentahalides are sparingly soluble in non-coordinating solvents, and they are easily susceptible to hydrolysis by traces of moisture, which make it troublesome to handle and store. This is a probable reason why the coordination chemistry of these group 5 metals is significantly less developed than that of other transition metal halides (e.g., group 4 metal tetrahalides), and the data reported in the literature up to 2007 were sparse and incomplete [11]. Moreover, the *reactivity* of niobium and tantalum pentahalides with limited, stoichiometric amounts of oxygen donor ligands had been scarcely investigated in the past, with only a few coordination adducts of $MCl_5$ (M=Nb, Ta) with O donors reported [12]. Kinetic investigations of niobium(V) and tantalum(V) complexes are quite challenging and are still incomplete [9–12] due to the tendency and ease of hydrolysis of ligands from these two hard metals; some results as catalysts have been presented, which show promise for future application [13,14].

This paper expands the range of O,O'-bidentate ligands with the target candidate 2,5-dichloro-3,6-dihydroxy-1,4-benzoquinone (chloranilic acid, $caH_2$) (Scheme 1) as a subclass of quinoid compounds, which are promising for the synthesis of novel functional materials. It is considered a unique multifunctional ligand system because these ligands possess coordination, hydrogen bonding and ionic interaction sites, together with redox active $\pi$-electronic structures affording rich coordination chemistry while also acting as strong proton donors and acceptors [15,16]. In the case of hydrogen-bond-supported supramolecular compounds, the flexibility in proton donation and/or acceptance of the donor and acceptor allows the formation of new compounds with well-designed geometries [17–19]. The characterization of these quinone-type of complexes is very important for the identification and prediction of its activity and selectivity in various reactions as well as in discovering significant chemical differences that could potentially be used for the separation of niobium from tantalum. There is also a search for complexes that will be attractive and useful as advanced materials for high-technology applications and in the nuclear power environment. The effective bridging function of $\mu$-chloranilate and the long-distance antiferromagnetic exchange interactions of binuclear copper(II) complexes containing the $\mu$-chloranilate ligand have been illustrated by single-crystal X-ray and magnetic analyses [20–22]. However, to our knowledge, no binuclear Nb(V) complex bridged by a chloranilate group of this kind has been reported yet. In addition, the fact that the $caH_2$ ligand, upon successive deprotonation, can 're-adjust' the delocalization and can behave in a '*cis*-deprotonation' fashion, as evidenced in **5**, further highlights the versatility of this ligand system.

Given the above, it is critical to increase our knowledge and potentially improve the stability of the various niobium(V) and tantalum(V) complexes for further application. Due to the unstable (e.g., unreactive and hydrolysis) nature of the general synthon niobium pentachloride, $NbCl_5$, tetraethylammonium chloride, $Et_4NCl$ can be used as a counter ion to generate tetraethylammonium hexachloridoniobate(V), $(Et_4N)[NbCl_6]$ (**1**). Here, we report its use and the characterization of the stable tetranuclear cluster obtained by self-assembly, and mono-nuclear complex, as well as substitution kinetics on the latter to contribute further to the *reactivity* knowledge base of niobium(V) complexes, which is crucial to furthering the potential development of separation processes on an industrial level. Hence, we also present a detailed kinetic investigation [23,24] of the substitution reaction between *cis*-$[NbO(ca)_2(H_2O)OPPh_3]^-$ and pyridine derivatives (Py) as entering nucleophiles, underlining the subtle interplay between hard transition metal complexes and main group ligand architectures.

**Scheme 1.** Dissociation of chloranilic acid (caH$_2$) to the monoanion (caH$^-$) and dianions (ca$^{2-}$), with resonance structure(s) (p$K_{a_1}$ = 0.76 and p$K_{a_2}$ = 2.58) [25].

## 2. Results

### 2.1. Synthesis of Compounds

2,5-Dichloro-3,6-dihydroxy-1,4-benzoquinone, caH$_2$, is a strong dibasic acid which possesses both electron-accepting and proton-donating properties and undergoes multistage deprotonation processes [26,27]. Hydrogen-bonding and ionic interaction sites were also employed as an important element for supramolecular architecture and crystal engineering [2,28].

The compounds (Et$_4$N)*cis*-[NbO(ca)$_2$(H$_2$O)OPPh$_3$]·3H$_2$O·THF (**5**) and (Et$_4$N)$_4$[Nb$_4$O$_4$ (ca)$_2$($\mu^2$-O)$_2$-Cl$_8$]·2CH$_3$CN (**6**) were synthesized and obtained as described in the experimental section (illustrated in Schemes 2 and 3). These show the direct two-step reactions at atmospheric conditions, resulting in a high percentage yield. Firstly, the starting material niobium pentachloride, NbCl$_5$, reacts with tetraethylammonium chloride (used as a counter ion) to yield tetraethylammonium hexachloridoniobate(V), (Et$_4$N)[NbCl$_6$] (Scheme 2, **1**) in an overall yield of 94.3%. In the second step, the successive proton dissociation of chloranilic acid (caH$_2$) generates the mono-(caH$^-$) (**3**) (Schemes 1 and 2) and dianion (ca$^{2-}$) (**4**) (Schemes 1 and 3), using triethylamine in 1:2 and 2:1 (caH$_2$:Et$_3$N) ratios, respectively. To obtain complexes **5** and **6**, ligand **3** (Scheme 1) was added to an acetonitrile solution of (Et$_4$N)[NbCl$_6$] (**1**), which gives the products **5** and **6** in overall yields of 81.7% and 85.6%, respectively.

**Scheme 2.** Synthesis of (Et$_4$N)*cis*-[NbO(ca)$_2$(H$_2$O)OPPh$_3$]·3H$_2$O·THF (**5**). Counter ions/solvates are omitted for clarity.

**Scheme 3.** Synthesis of $(Et_4N)_4[Nb_4O_4(ca)_2(\mu^2\text{-}O)_2Cl_8]\cdot 2CH_3CN$ (**6**). Counter ions/solvates are omitted for clarity.

All complexes could be well characterized by IR, UV/Vis and ($^1$H-, $^{13}$C{$^1$H}- and $^{31}$P{$^1$H}) NMR spectroscopy as well as X-ray diffraction analysis as reported in the experimental section.

### 2.2. X-ray Crystallography

The crystal structure of 2,5-dichloro-3,6-dihydroxy-2,5-cyclohexadiene-1,4-dione, caH$_2$, was first described by Andersen in 1967 [29] and was collected at room temperature using visually estimated intensities of the diffraction spots obtained by means of the Weissenberg equi-inclination method. Later, this structure was reported by Dutkiewicz et al., in 2010 [30] at low temperature, describing the structure in the $P2_1/n$ space group instead of the $P2_1/a$ used by Andersen. Interestingly, in this study, we obtained the caH$_2$ as a water solvate in the $P2_1/c$ space group with different cell parameters. This free ligand structure, (caH$_2$)·2H$_2$O (**2**) presented herein was thus redetermined at 100 (2) K (Figure 1) and lies on an inversion center.

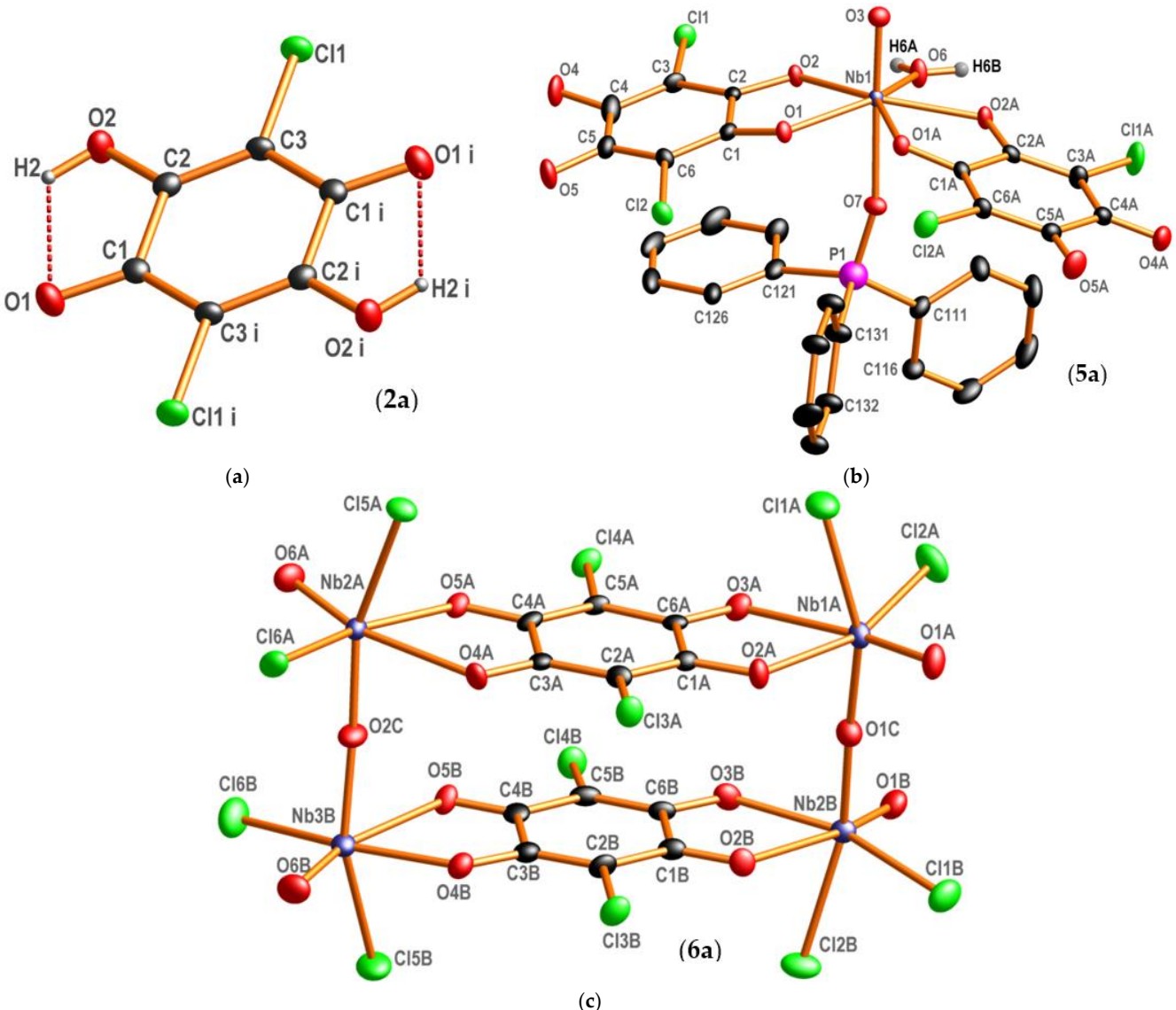

**Figure 1.** Perspective view of (**a**) (caH$_2$) (**2a**); atoms generated by symmetry are indicated by lower case roman numerals corresponding to the symmetry operator (i) 1−x, −y, −z; (**b**) *cis*-[NbO(ca)$_2$(H$_2$O)OPPh$_3$]$^-$ (**5a**); and (**c**) [Nb$_4$O$_4$(ca)$_2$($\mu^2$-O)$_2$Cl$_8$]$^{4-}$ (**6a**). Ellipsoids are drawn at a 50% probability. Hydrogen atoms, counter ions, and solvate molecules are omitted for clarity. Carbon atoms in counter ions in **5** and **6**, as well as the phenyl rings of the OPPh$_3$ in **5**, show typical small dynamic disorders which do not significantly influence the coordination geometries observed, and they were not solved.

This study forms part of ongoing research to investigate the mechanism of the reactions of O,O'-, N,O-, and O,S-bidentate ligands with transition metals used in the nuclear industry, specifically hafnium, zirconium, niobium and tantalum [31,32]. The free ligand, (caH$_2$)·2H$_2$O (**2**) and coordination compounds (Et$_4$N)*cis*-[NbO(ca)$_2$(H$_2$O)-OPPh$_3$]·3H$_2$O·THF (**5**) and (Et$_4$N)$_4$[Nb$_4$O$_4$(ca)$_2$($\mu^2$-O)$_2$Cl$_8$]·2CH$_3$CN (**6**) crystallizes in the monoclinic space group, *P*2$_1$/*c* with Z = 2, *P*2$_1$ with Z = 2 and *P*2$_1$/*c* with Z = 4, respectively. Table 1 summarizes the crystallographic data and refinement parameters of **5** and **6** as well as the free ligand, **2**, while Tables 2–4 list, compare and correlate different structural parameters within **2a**, **5a** and **6a**.

**Table 1.** Crystallographic data for $(caH_2) \cdot 2H_2O$ (**2**), $(Et_4N)cis$-$[NbO(ca)_2(H_2O)OPPh_3] \cdot 3H_2O \cdot THF$ (**5**) and $(Et_4N)_4[Nb_4O_4(ca)_2(\mu^2\text{-}O)_2Cl_8] \cdot 2CH_3CN$ (**6**).

| Identification Code | 2 | 5 | 6 |
|---|---|---|---|
| Empirical formula | $C_6H_6Cl_2O_6$ | $C_{42}H_{51}Cl_4NNbO_{15}P$ | $C_{48}H_{86}Cl_{12}N_6Nb_4O_{14}$ |
| Formula weight | 122.5 | 1075.5 | 1768.3 |
| Temperature (K) | 100 (2) | 100 (2) | 100 (2) |
| Wavelength (Å) | 0.71073 | 0.71073 | 0.71073 |
| Crystal system, space group | Monoclinic, $P2_1/c$ | Monoclinic, $P2_1$ | Monoclinic, $P2_1/c$ |
| $a$ (Å) | 8.5571 (15) | 10.468 (7) | 21.598 (4) |
| $b$ (Å) | 10.3147 (16) | 18.005 (13) | 10.666 (2) |
| $c$ (Å) | 5.135 | 12.584 (9) | 31.201 (6) |
| $\alpha$ (°) | 90.000 | 90.00 | 90.00 |
| $\beta$ (°) | 104.420 (6) | 92.66 (3) | 92.198 (10) |
| $\gamma$ (°) | 90.000 | 90.00 | 90.00 |
| Volume (Å$^3$) | 438.97 (10) | 2369.2 (3) | 7182 (2) |
| Z | 4 | 2 | 4 |
| Density$_{calc}$ (g.cm$^{-3}$) | 1.854 | 1.508 | 1.6326 |
| μ (mm$^{-1}$) | 0.740 | 0.580 | 1.127 |
| Crystal size (mm) | $0.119 \times 0.329 \times 0.448$ | $0.143 \times 0.320 \times 0.788$ | $0.6 \times 0.46 \times 0.17$ |
| R$_{int}$ | 0.0375 | 0.0458 | 0.0710 |
| Goodness-of-fit on F$^2$ | 1.194 | 1.026 | 1.201 |
| Final $R$ indices ($I > 2\sigma(I)$) | $R1 = 0.0205$ $wR2 = 0.0566$ | $R1 = 0.0341$ $wR2 = 0.0849$ | $R1 = 0.1117$ $wR2 = 0.2600$ |
| $\Delta\rho_{max}$ and $\Delta\rho_{min}$ (e.Å$^{-3}$) | 0.338, −0.278 | 1.522, −1.353 | 1.896, −3.264 |

R1 = $\Sigma||F_o| - |F_c||/\Sigma|F_o|$, wR2 = $[\Sigma[w(F_o{}^2 - F_c{}^2)^2]/\Sigma[w(F_o{}^2)^2]]^{1/2}$ and GOF = $[\Sigma[w(F_o{}^2 - F_c{}^2)^2]/(M - N)]^{1/2}$, where M is the number of reflections and N is the number of parameters refined.

**Table 2.** Selected interatomic bond lengths and bond angles within the chloranilic acid/chloroanilate ligand in $(caH_2) \cdot 2H_2O$ (**2**), $(Et_4N)cis$-$[NbO(ca)_2(H_2O)OPPh_3] \cdot 3H_2O \cdot THF$ (**5**) and $(Et_4N)_4[Nb_4O_4(ca)_2(\mu^2\text{-}O)_2Cl_8] \cdot 2CH_3CN$ (**6**).

| Bond Lengths (Å) | | | | | | |
|---|---|---|---|---|---|---|
| Type of Bond | 2 | (Å) | 5 | (Å) | 6 | (Å) |
| C=O | C1-O1 | 1.2256 (15) | C1-O1 | 1.316 (3) | C6b-O3b | 1.269 (12) |
|  | C1i-O1i | 1.2256 (15) | C2-O2 | 1.314 (3) | C1b-O2b | 1.237 (12) |
|  |  |  | C1a-O1a | 1.278 (5) | C4b-O5b | 1.248 (12) |
|  |  |  | C2a-O2a | 1.288 (5) | C3b-O4b | 1.279 (12) |
|  | Average | 1.226 |  | 1.299 |  | 1.258 |
| C-O | C2-O2 | 1.3173 (14) | C4-O4 | 1.301 (4) | C6a-O3a | 1.241 (12) |
|  | C2i-O2i | 1.3173 (14) | C5-O5 | 1.285 (4) | C1a-O2a | 1.278 (11) |
|  |  |  | C4a-O4a | 1.227 (5) | C4a-O5a | 1.278 (12) |
|  |  |  | C5a-O5a | 1.224 (5) | C3a-O4a | 1.253 (12) |
|  | Average | 1.317 |  | 1.259 |  | 1.263 |
| C-Cl | C3-Cl1 | 1.7186 (12) | C3-Cl1 | 1.799 (2) | C2b-Cl3b | 1.747 (10) |
|  | C3i-Cl1i | 1.7186 (12) | C3a-Cl1a | 1.725 (5) | C5b-Cl4b | 1.724 (10) |
|  |  |  | C6-Cl2 | 1.782 (2) | C2a-Cl3a | 1.713 (10) |
|  |  |  | C6a-Cl2a | 1.728 (4) | C5a-Cl4a | 1.728 (9) |
|  | Average | 1.717 |  | 1.759 |  | 1.728 |
| Bond angles (°) | | | | | | |
| Type of Angle | 2 [a] | (°) [a] | 5 [b] | (°) | 6 [b] | (°) |
| O-(C-C)$_{MC}$ [c] | O1 C1 C2 | 118.15 (11) | $O_{k1}$-$C_{k2}$-$C_{k3}$ [d] | 115.4 | $O_{k1}$-$C_{k2}$-$C_{k3}$ [d] | 115.4 |
| O-C-**C**$_{Out}$ [e] | O2 C2 C3 | 122.05 (11) | $O_{l1}$-$C_{l2}$-$C_{l3}$ [f] | 123.7 | $O_{l1}$-$C_{l2}$-$C_{l3}$ [f] | 125.3 |
| C-C-**Cl**$_{Out}$ [g] | C2 C3 Cl1 | 121.12 (9) | $C_{m1}$-$C_{m2}$-$Cl_{m3}$ [h] | 120.0 | $C_{m1}$-$C_{m2}$-$Cl_{m3}$ [h] | 119.4 |
| C-C-C [i] | C3 C2 C1 | 120.18 (10) | $C_{n1}$-$C_{n2}$-$C_{n3}$ [j] | 120.0 | $C_{n1}$-$C_{n2}$-$C_{n3}$ [j] | 120.0 |

[a] Free ligand included for comparison; [b] Average of a number of similar values; esds of average values for **5** < 0.3°; and for **6** < 1°; [c] (C-C)$_{MC}$ = carbon atoms defining the metallocyle; [d] k1 (oxygen), k2 (carbon), k3 (carbon); [e] **C**$_{Out}$ = 'outer' carbon atoms of ca-ring; [f] l1 (oxygen), l2 (carbon), l3 (carbon); [g] **Cl**$_{Out}$ = chlorine atoms on ca-ring; [h] m1 (carbon), m2 (carbon), m3 (chlorine); [i] Carbon atoms in ca-ring; [j] n1, n2, n3: carbon atoms. All of these are also defined in Supplementary Materials, Table S1.

**Table 3.** Selected coordination bond lengths in $(Et_4N)cis$-[$NbO(ca)_2(H_2O)OPPh_3$]·$3H_2O$·THF (**5**) and $(Et_4N)_4[Nb_4O_4(ca)_2(\mu^2$-$O)_2Cl_8]$·$2CH_3CN$ (**6**).

| Bond Lengths | 5 | | 6 | | | |
|---|---|---|---|---|---|---|
| Type Bond | Bond | (Å) | Bond | (Å) | Bond | (Å) |
| Nb-O$_{ca}$ | Nb1-O1 | 2.103 (3) | Nb3b-O4b | 2.142 (7) | Nb2a-O5a | 2.129 (7) |
| | Nb1-O2 | 2.116 (3) | Nb2b-O3b | 2.132 (7) | Nb1a-O2a | 2.139 (7) |
| | Nb1-O1a | 2.149 (3) | Nb3b-O5b | 2.402 (7) | Nb2a-O4a | 2.398 (7) |
| | Nb1-O2a | 2.145 (3) | Nb2b-O2b | 2.390 (7) | Nb1a-O3a | 2.381 (7) |
| Nb-O$_{OPPh3}$ | Nb1-O7 | 2.192 (3) | - | - | - | - |
| Nb-OH$_2$ | Nb1-O6 | 2.108 (3) | - | - | - | - |
| Nb-O-Nb | - | - | Nb2b-O1c | 1.900 (8) | Nb2a-O2c | 1.896 (7) |
| | - | - | Nb1a-O1c | 1.919 (8) | Nb3b-O2c | 1.914 (7) |
| Nb=O | Nb1-O3 | 1.718 (3) | Nb3b-O6b | 1.716 (8) | Nb2a-O6a | 1.731 (8) |
| | - | - | Nb2b-O1b | 1.705 (8) | Nb1a-O1a | 1.711 (8) |
| Nb-Cl | - | - | Nb2b-Cl1b | 2.383 (3) | Nb1a-Cl1a | 2.464 (3) |
| | - | - | Nb2b-Cl2b | 2.469 (3) | Nb1a-Cl2a | 2.376 (3) |
| | - | - | Nb3b-Cl6b | 2.375 (3) | Nb2a-Cl5a | 2.459 (3) |
| | - | - | Nb3b-Cl5b | 2.450 (3) | Nb2a-Cl6a | 2.386 (3) |
| Nb-Nb | - | - | Nb2b-Nb2a | 9.131 (2) | Nb2b-Nb3b | 8.406 (2) |
| | - | - | Nb3b-Nb1a | 9.265 (2) | Nb1a-Nb2a | 8.375 (2) |
| | - | - | Nb2b-Nb1a | 3.818 (6) | Nb3b-Nb2a | 3.807 (6) |

**Table 4.** Selected bond angles in $(Et_4N)cis$-[$NbO(ca)_2(H_2O)OPPh_3$]·$3H_2O$·THF (**5**) and $(Et_4N)_4[Nb_4O_4(ca)_2(\mu^2$-$O)_2Cl_8]$·$2CH_3CN$ (**6**).

| Bond Angles | 5 | | 6 | | | |
|---|---|---|---|---|---|---|
| Type Angle | Angle | (°) | Angle | (°) | Angle | (°) |
| (O-Nb-O)$_{bite}$ | O1-Nb1-O2 | 73.07 (11) | O3b-Nb2b-O2b | 70.0 (3) | O3a-Nb1a-O2a | 70.9 (3) |
| | O1a-Nb1-O2a | 71.93 (11) | O5b-Nb3b-O4b | 69.8 (3) | O5a-Nb2a-O4a | 70.3 (2) |
| | O1a-Nb1-O1 | 72.07 (11) | - | - | - | - |
| | O2-Nb1-O6 | 70.52 (11) | - | - | - | - |
| O=Nb-O$_{ca}$ | O3-Nb1-O1 | 98.46 (13) | O1b-Nb2b-O3b | 95.0 (3) | O1a-Nb1a-O2a | 97.6 (3) |
| | O1a-Nb1-O3 | 92.39 (13) | O6b-Nb3b-O4b | 94.0 (4) | O6a-Nb2a-O5a | 94.1 (3) |
| | O6-Nb1-O7 | 86.89 (12) | O1b-Nb2b-O1c | 100.3 (3) | O1a-Nb1a-O1c | 100.9 (4) |
| | O2-Nb1-O7 | 89.47 (12) | O6b-Nb3b-O2c | 102.1 (4) | O6a-Nb2a-O2c | 103.1 (4) |
| O=Nb-O$_{trans}$ | O2a-Nb1-O2 | 140.32 (12) | O1b-Nb2b-O2b | 164.8 (3) | O1a-Nb1a-O3a | 168.4 (3) |
| | O3-Nb1-O7 | 174.93 (13) | O6b-Nb3b-O5b | 163.6 (3) | O6a-Nb2a-O4a | 163.4 (3) |
| Cl-Nb-Cl | - | - | Cl1b-Nb2b-Cl2b | 88.12 (11) | Cl6b-Nb3b-Cl5b | 89.36 (12) |
| | - | - | Cl2a-Nb1a-Cl1a | 88.97 (12) | Cl6a-Nb2a-Cl5a | 88.49 (10) |
| O=Nb-Cl | - | - | O1b-Nb2b-Cl2b | 96.1 (3) | O1b-Nb2b-Cl1b | 103.0 (3) |
| | - | - | O6b-Nb3b-Cl5b | 99.5 (3) | O6b-Nb3b-Cl6b | 100.5 (3) |
| | - | - | O1a-Nb1a-Cl1a | 97.5 (3) | O1a-Nb1a-Cl2a | 102.5 (3) |
| | - | - | O6a-Nb2a-Cl5a | 96.1 (3) | O6a-Nb2a-Cl6a | 102.5 (3) |

As indicated, Tables 2–4 list and compare different bond distances and – angles in **2**, **5** and **6**, which are discussed in Section 3.2.

### 2.3. Kinetic Study of the OPPh₃ Substitution from cis-[NbO(ca)₂(H₂O)OPPh₃]⁻ (5a) by Py-Type Nucleophiles

The substitution of triphenylphosphine oxide from **5a** by different pyridine derivatives was studied by time-resolved spectroscopy (UV/Vis, NMR and IR). Ligand substitution reactions on the seven-coordinate metal complex *cis*-[NbO(ca)₂(H₂O)OPPh₃]⁻ (**5a**) have been successfully studied using a pre-selected range of pyridine-type derivatives (Py) as a monodentate entering nucleophiles/ligands. The pyridine derivatives which were selected for these kinetic studies span a significant range of electron-donating abilities while keeping the steric demand of the entering nucleophile essentially constant (Section 2.3.5; Table 5). These are pyridine (py), 4-(dimethylamino)pyridine (DMAP), 4-methylpyridine (4Mepy), 3-chloropyridine (3Clpy) and 3-bromopyridine (3Brpy) (Supplementary Materials, Tables S19 and S20). The results of these studies almost universally suggested an associative mechanism [24]. Note that since DMAP formed the most stable product, it was selected for temperature and other more detailed experiments, as described in Sections 2.3.1–2.3.6, rather than the other Py-type nucleophiles to ensure better conversions to the final product.

**Table 5.** Equilibrium, p$K_a$, and second-order rate constants (Equations (3) and (4)) for the substitution of triphenylphosphine oxide from *cis*-[NbO(ca)₂(H₂O)OPPh₃]⁻ by different entering Py-type nucleophiles in acetonitrile at 31.2 °C.

| Entering Py Nucleophiles | | p$K_a$ [a] | $K_{eq}$ [b] | $k_f$ | $k_r$ |
|---|---|---|---|---|---|
| Number | Name | | | $(M^{-1}s^{-1}) \times 10^{-2}$ [c] | $(s^{-1}) \times 10^{-2}$ [c] |
| 1 | DMAP | 9.60 | 24 ± 5 | 2.03 ± 0.05 | 0.08 ± 0.02 |
| 2 | 4Mepy | 6.02 | 1.91 ± 0.08 | 0.180 ± 0.006 | 0.09 ± 0.02 |
| 3 | py | 5.25 | 1.09 ± 0.02 | 0.130 ± 0.001 | 0.12 ± 0.01 |
| 4 | 3Clpy | 2.84 | 2.82 ± 0.13 | 0.210 ± 0.007 | 0.07 ± 0.02 |
| 5 | 3Brpy | 2.84 | 2.15 ± 0.09 | 0.150 ± 0.005 | 0.07 ± 0.01 |

[a] [25]; [b] Equation (4); [c] Equation (3).

Unfortunately, the final substitution reaction product, *cis*-[NbO(ca)₂(H₂O)(DMAP)]⁻, could not be crystallized for analysis by single crystal X-ray diffraction nor isolated in pure form, and an expanded attempt using spectroscopy was therefore necessitated and attempted. Before this, however, exhaustive yet unsuccessful attempts to try to crystallize all five pyridine derivative complexes which were used for the kinetic study, by using the same procedure from which the starting complex crystallize structure, were launched. This effort, as well as using different crystallization techniques unfortunately did not yield any crystalline substitution products which could be analyzed further.

Moreover, because the reactions are clear equilibria (see Section 2.3.4), an excess of the pyridine ligands was always required, which contaminated the products. As soon as crystallization of the products was attempted, hydrolysis of the pyridine-type ligand was observed. Nevertheless, the kinetic analyses were in our view convincingly performed and corroborated the fact that only the triphenylphosphine oxide was substituted.

#### 2.3.1. UV/Vis Measurements

The substitution of the triphenylphosphine oxide by DMAP was monitored utilizing the absorbance change at a suitable fixed wavelength of 430 nm for the complex, *cis*-[NbO(ca)₂(H₂O)OPPh₃]⁻ (**5a**), as selected from pre-recorded absorbance versus wavelength spectra (Figure 2). All the pyridine ligand concentrations were varied between 0.1 and 1.0 M, whereas a concentration of 0.001 M for the *cis*-[NbO(ca)₂(H₂O)OPPh₃]⁻ (**5a**) was used. A stock solution of 1.0 M pyridine was prepared and diluted to obtain the appropriate concentrations for the kinetic runs in acetonitrile. The metal complex and ligand stability of solutions in the reaction solvent, acetonitrile (MeCN), was confirmed over a ca. 24 h period, for each of the five pyridine derivative ligands studied.

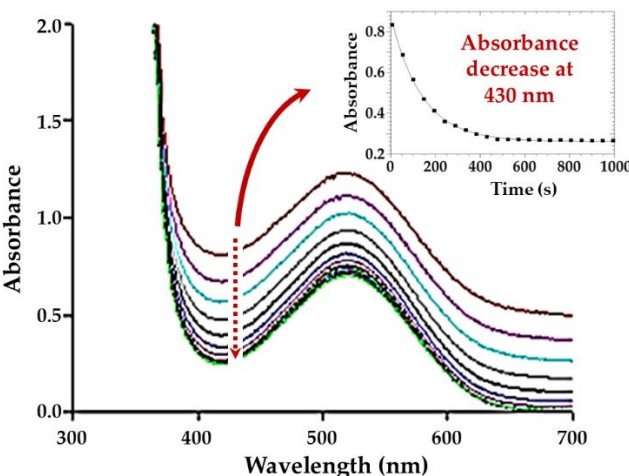

**Figure 2.** UV/Vis spectral change for the substitution of triphenylphosphine oxide in *cis*-[NbO(ca)$_2$(H$_2$O)OPPh$_3$]$^-$ (**5a**) ([0.001 M]) by DMAP ([0.35 M]) at 31.2 °C, 430 nm in acetonitrile; $\Delta t$ = 50 s, total time of 1000 s. The insert indicates the absorbance change versus time at 430 nm, and the solid line shows the least-squares fit as described previously [23,24] for the first-order reaction ($k_{obs}$ = (7.6 ± 0.2) × 10$^{-3}$ s$^{-1}$).

The rate constants for the substitution of triphenylphosphine oxide by the DMAP ligand were thus obtained by monitoring the absorbance changes at 430 nm under *pseudo* first-order conditions, which indicated that there is only one reaction taking place, with no evidence of neither a fast pre-reaction nor a second slower reaction step present (Figure 2) [33–35]. The least-squares fitting of the data sets was performed as previously described [23,24]. Acetonitrile was the only solvent in which the starting complex is quite stable, and it gave repeatable kinetic results. However, during the NMR ($^1$H and $^{31}$P) experiment, the solubility of the starting complex, *cis*-[NbO(ca)$_2$(H$_2$O)OPPh$_3$]$^-$, was not complete in acetonitrile due to the high concentrations required for acceptable peak interpretation, but for the DMAP ligand, the solubility was quite good.

In another preliminary experiment, triphenylphosphine oxide was added to evaluate if an excess in solution would influence the reaction, but no significant change in the rates was observed.

The $^{31}$P{$^1$H} NMR study described in Section 2.3.2 addresses the question of the formation of *cis*-[NbO(ca)$_2$(<u>MeCN</u>)OPPh$_3$]$^-$, following dissolution in a coordinating solvent such as acetonitrile. Even in virtually pure MeCN, only some 30% of complex **4a** is converted to the corresponding MeCN complex. It is clear from the results that no deviation from first-order kinetics (Figure 2) was observed, and it was therefore concluded that the presence of (some of) the MeCN reactant did not influence the reaction significantly.

### 2.3.2. $^{31}$P NMR{$^1$H} Study

A preliminary $^{31}$P{$^1$H} NMR experiment was performed to determine whether hydrolysis of the triphenylphosphine from **5a** was realistic. The results as illustrated in Figure 3 underline complicating factors in the substitution process. Therein, free OPPh$_3$ with excess water (0.017 M) yields a signal with a chemical shift at 29.6 ppm. Moreover, after the addition of a 1.0:2.2 molar ratio of starting complex (**a**) to DMAP, the liberated free OPPh$_3$ signal appears at a chemical shift of 29.6 ppm, which means the *cis* aqua ligand in the reactant *cis*-[NbO(ca)$_2$(H$_2$O)OPPh$_3$]$^-$ species is not substituted by DMAP, only the OPPh$_3$. Thus, from this, it is concluded that the final product is *cis*-[NbO(ca)$_2$(H$_2$O)(DMAP)]$^-$. The signals at 45.9 and 38.9 ppm represent the coordinated OPPh$_3$ in *cis*-[NbO(ca)$_2$(MeCN)OPPh$_3$]$^-$ (33%) and *cis*-[NbO(ca)$_2$(H$_2$O)OPPh$_3$]$^-$ (67%) in the starting complex (**a**), respectively. Spectrum (**e**) represents the free OPPh$_3$ (signal at 29.6 ppm) upon the addition of excess water (0.017 M). Moreover, **b**, **c**, and (**d**) show that the *cis*-[NbO(ca)$_2$(MeCN)OPPh$_3$]$^-$ (**Y**) species disappeared and the average signal at 38.9 ppm represents the coordinated OPPh$_3$ in the

*cis*-[NbO(ca)$_2$(H$_2$O)OPPh$_3$]$^-$ (**Z**) as well as the average signal at 29.6 ppm, a new species which represents free OPPh$_3$, after the addition of excess water (0.002, 0.004, and 0.010 M) to the starting complex (**a**), respectively. The additional conclusion from this experiment is that a large amount of water can liberate the OPPh$_3$ from the starting complex (**a**).

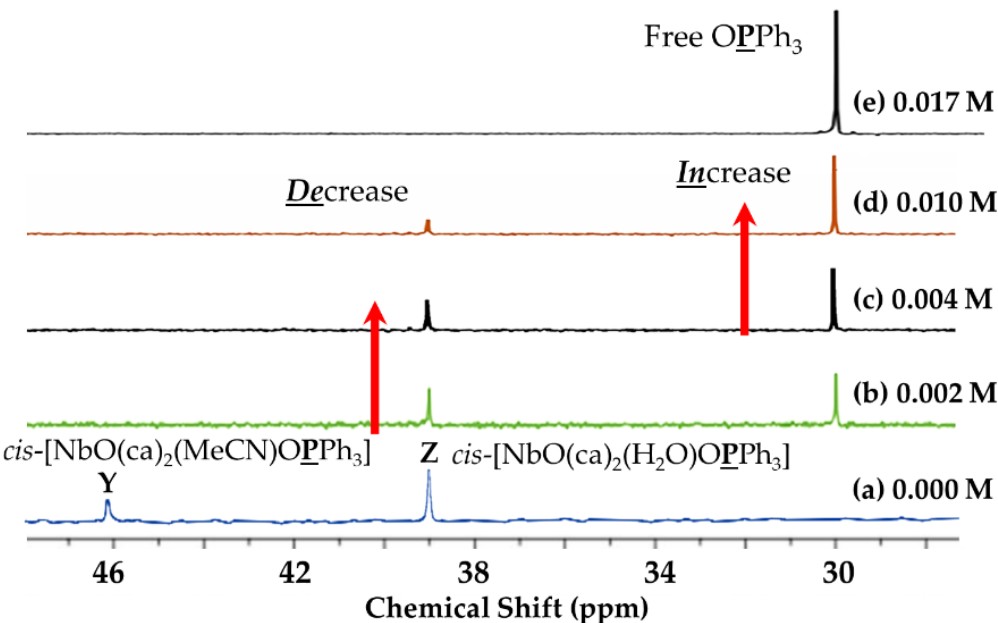

**Figure 3.** Stacked plot of $^{31}$P{$^1$H} NMR spectral change which is influenced by the addition of water to the starting complex (**a**), *cis*-[NbO(ca)$_2$(H$_2$O)OPPh$_3$]$^-$ (0.013 M), ((**b–e**), 0.002, 0.004, 0.010 and 0.017 M), respectively in acetonitrile (CD$_3$CN) at 25 °C, showing the decreasing *cis*-[NbO(ca)$_2$(MeCN)OPPh$_3$]$^-$ (**Y**), and *cis*-[NbO(ca)$_2$(H$_2$O)OPPh$_3$]$^-$ (**Z**), and increasing liberated free OPPh$_3$ (signal at 29.9 ppm).

In another preliminary evaluation to confirm that OPPh$_3$ is (or not) substituted by DMAP, a second $^{31}$P{$^1$H} NMR study was undertaken. The experiment involved the dissolution of the starting complex, (Et$_4$N)*cis*-[NbO(ca)$_2$(H$_2$O)OPPh$_3$], 0.013 M (7.5 mg in 600 µL deuterated acetonitrile), which was followed by the successive addition of a range of concentrations of DMAP (between 0.013 and 0.030 M) as illustrated in Figure 4. The signals at 46.0 and 39.5 ppm represent coordinated triphenylphosphine oxide, of which 33% of the total is assumed to be the *cis*-[NbO(ca)$_2$(MeCN)OPPh$_3$]$^-$ and 67% is *cis*-[NbO(ca)$_2$(H$_2$O)OPPh$_3$]$^-$ in the starting complex (**a**), respectively. Here, it is concluded that there is an equilibrium between the MeCN and H$_2$O species. Spectra **b**, **c**, **d**, **e** and **f**, on the other hand, illustrate the stepwise substitution of OPPh$_3$ by DMAP (0.005, 0.010, 0.015, 0.020, and 0.025 M), respectively. Moreover, **g** represents the complete substitution reaction between *cis*-[NbO(ca)$_2$(H$_2$O)OPPh$_3$]$^-$ and DMAP and gives a new product, *cis*-[NbO(ca)$_2$(H$_2$O)(DMAP)]$^-$ as well as liberated free OPPh$_3$ in CD$_3$CN at 25 °C. The overall conclusion from this experiment is that the OPPh$_3$ is being substituted in an equilibrium reaction to yield the *cis*-[NbO(ca)$_2$(H$_2$O)(DMAP)]$^-$ complex, as further elaborated in Section 2.3.4.

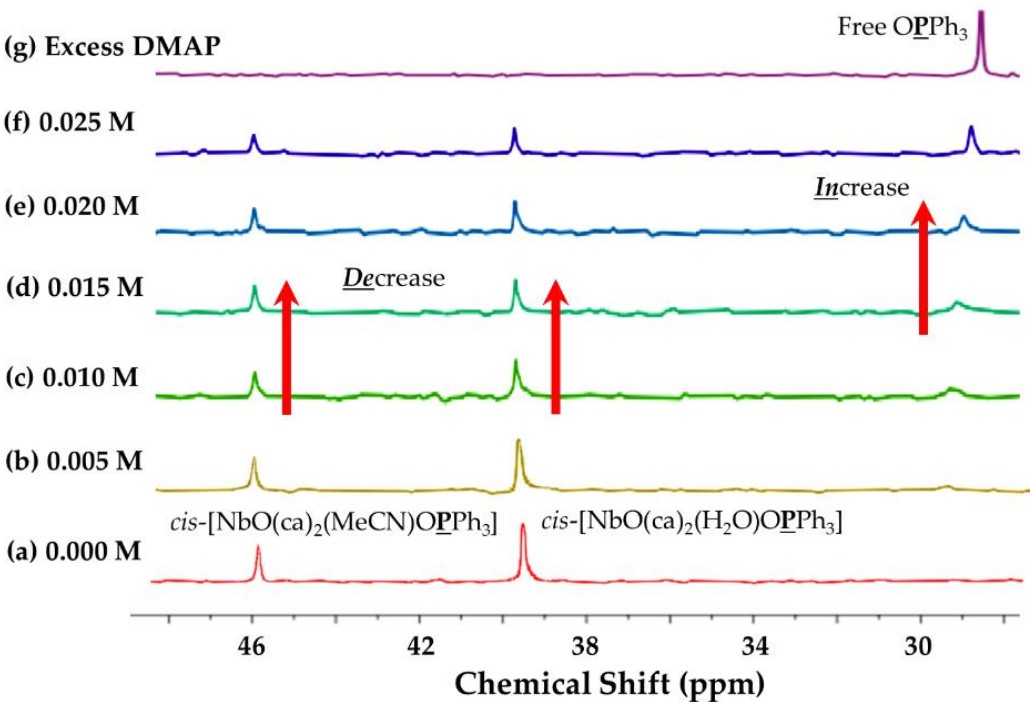

**Figure 4.** Illustration of $^{31}$P{$^1$H} NMR spectral change (after equilibrium has been reached) for the substitution of triphenylphosphine oxide from *cis*-[NbO(ca)$_2$(H$_2$O)OPPh$_3$]$^-$ (0.013 M) (**a**) by DMAP (0.005–0.03 M) (**b**–**g**). Illustrated is the decrease in both the *cis*-[NbO(ca)$_2$(MeCN)OPPh$_3$]$^-$ and the *cis*-[NbO(ca)$_2$(H$_2$O)OPPh$_3$]$^-$, accompanied by the increasing liberated free OPPh$_3$ product the signal at 29.6 ppm, at 25 °C in CD$_3$CN. Time allowed after each addition = 30 min.

2.3.3. $^1$H NMR Studies

Next, a $^1$H NMR evaluation was performed to attempt to evaluate the potential influence of the addition of DMAP to the system and the effects of (i) concentrations and (ii) temperature on the reaction.

Concentration Effect of DMAP

Figure 5 illustrates the $^1$H NMR spectra showing the concentration dependence of DMAP as the entering ligand on the reaction and the changes that are observed in the OPPh$_3$ substitution from *cis*-[NbO(ca)$_2$(H$_2$O)OPPh$_3$]$^-$ (**a**) (0.013 M) by DMAP (0.013–0.068 M) (**b**–**g**) at 25 °C in CD$_3$CN. This shows that OPPh$_3$ is substituted by DMAP at concentrations of 0.026 M (**c**), but successive changes in the spectra, assumed to be due to fast exchange, after the addition of an excess of DMAP in the range between 0.03–0.068 M takes place. Thus, the spectral results also show some fast exchange and the dynamic nature of the process.

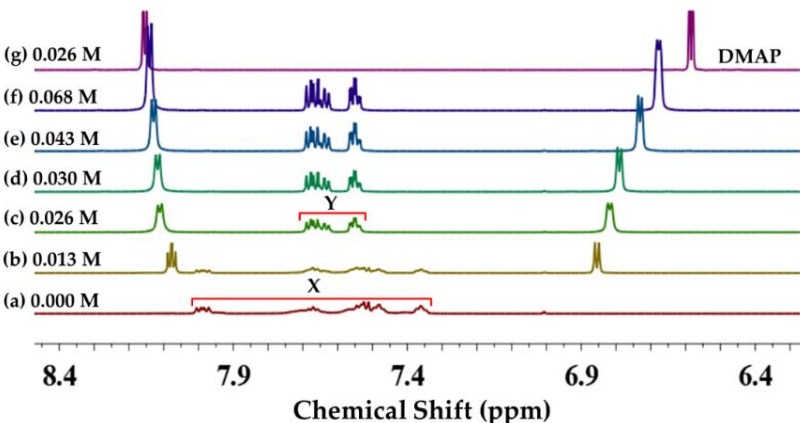

**Figure 5.** $^1$H NMR spectra illustrating the changes observed in the substitution of triphenylphosphine oxide from *cis*-[NbO(ca)$_2$(H$_2$O)OPPh$_3$]$^-$ (0.013 M) (**a**), by DMAP (0.013–0.068 M) (**b–f**) at 25 °C in CD$_3$CN. **X** represents coordinated OPPh$_3$ in the starting complex (**a**) (0.013 M), while **Y** gives the stepwise liberated OPPh$_3$ upon addition of an excess of DMAP (**f**) (0.068 M), respectively. Spectrum (**g**) was obtained using pure DMAP.

Temperature Effect

Figure 6 illustrates that the *cis*-[NbO(ca)$_2$(H$_2$O)OPPh$_3$]$^-$ complex and equilibria are not significantly affected by temperature. Moreover, the shapes of the signals are all very similar and indicate either (i) a very fast exchange or (ii) no exchange at all, and that the slight signal shapes observed might be due to proton exchange and not substitution processes.

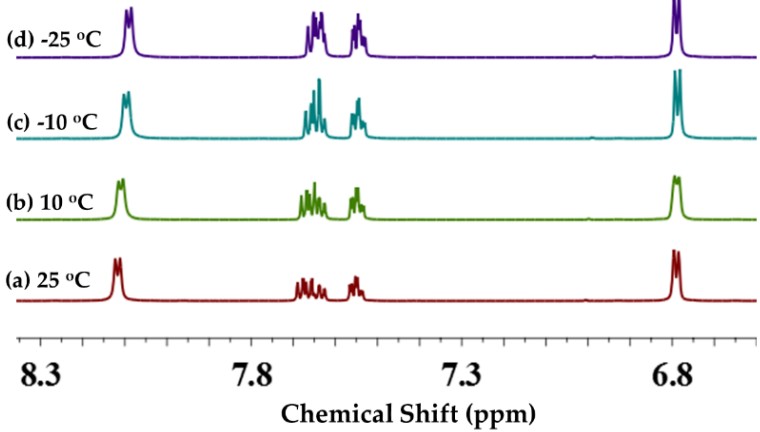

**Figure 6.** $^1$H NMR spectra showing the changes observed for the substitution of triphenylphosphine oxide from *cis*-[NbO(ca)$_2$(H$_2$O)OPPh$_3$]$^-$ (0.03 M) by DMAP (0.013 M) at four different temperatures ((**a–d**); 25, 10, −10, and −25 °C) in CD$_3$CN.

2.3.4. Mechanism of the Substitution of OPPh$_3$ from cis-[NbO(ca)$_2$(H$_2$O)OPPh$_3$]$^-$ by Pyridine-Type Ligands

The above results indicate that upon the addition of pyridine-type ligands to *cis*-[NbO(ca)$_2$(H$_2$O)OPPh$_3$]$^-$, only the OPPh$_3$ is substituted according to the reaction given in Equation (1), where $k_f$ and $k_r$ represent the forward and reverse rate constants for the simple ligation process, respectively [23,24].

$$cis-[NbO(ca)_2(H_2O)OPPh_3]^- + Py \underset{k_r}{\overset{k_f, K_1}{\rightleftharpoons}} cis-[NbO(ca)_2(H_2O)OPPh_3]^- + OPPh_3 \quad (1)$$

From Equation (1), the rate law to the process can be written as given in Equation (2).

$$\text{Rate} = -d[\text{NbP}]/dt = k_f[\text{NbP}][\text{Py}] - k_r[\text{NbPy}][\text{P}] \tag{2}$$

The NbP and NbPy represent the reactant and product species in Equation (1) (i.e., (**a**) and (**b**), respectively). By using conditions where [Py] >> [Nb] and following integration, the *pseudo* first-order rate constant shown in Equation (1) is given by Equation (3).

$$k_{obs} = k_f[\text{Py}] + k_r \tag{3}$$

From general mass balances and Le Chatelier's principle, it follows that

$$K_1 = k_f/k_r \tag{4}$$

A systematic study was therefore conducted wherein different pyridine-type ligands were utilized and a progressive increase in [Py] was studied. The results are presented in Section 2.3.5.

## 2.3.5. The Effect of Pyridine Concentration on OPPh$_3$ Substitution from *cis*-[NbO(ca)$_2$(H$_2$O)OPPh$_3$]$^-$ (**5a**)

In Equation (1), a general simple scheme is represented for the stoichiometric substitution [23,24] of OPPh$_3$ from *cis*-[NbO(ca)$_2$(H$_2$O)OPPh$_3$]$^-$ by pyridine-type ligands such as py, 4Mepy, DMAP, 3Clpy and 3Brpy. These were investigated in acetonitrile using UV/Vis spectrophotometry. Figure 7 illustrates a plot of the observed *pseudo* first-order rate constant, $k_{obs}$ versus Py-concentration. The substitution kinetics yielded straight lines with positive intercepts for all five Py-type ligands according to Equation (3).

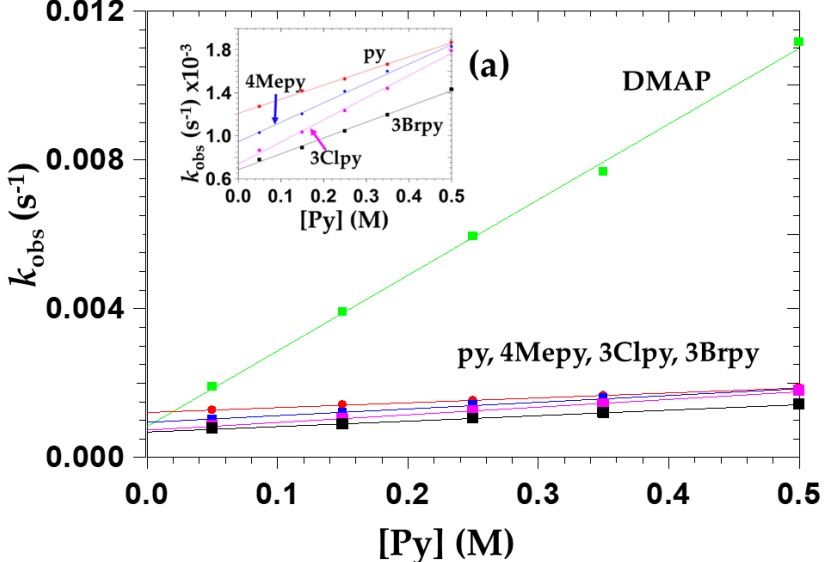

**Figure 7.** Systematic variation of the ligand concentration for the substitution of triphenylphosphine oxide from *cis*-[NbO(ca)$_2$(H$_2$O)OPPh$_3$]$^-$ by the pyridine-type ligands in acetonitrile at 31.2 °C. Insert (**a**) gives an enlarged perspective of the bottom four ligands. Ligand: green DMAP, red py, blue 4Mepy, purple 3Clpy, and black 3Brpy. Py = pyridine-type ligands.

Thus, Table 5 reports the comparison of the equilibrium constants $K_{eq}$, p$K_a$, and the rate constant values of the substitution of OPPh$_3$ by five different entering ligands from the starting complex, *cis*-[NbO(ca)$_2$(H$_2$O)OPPh$_3$]$^-$. The reaction rates with DMAP and 3Brpy are the highest and lowest observed compared to all five pyridine-type ligands, since DMAP and 3Brpy are more electron donating and withdrawing than the other listed ligands.

### 2.3.6. The Effect of Temperature on OPPh$_3$ Substitution from *cis*-[NbO(ca)$_2$(H$_2$O)OPPh$_3$]$^-$

The standard enthalpy change of activation ($\Delta H^{\neq}_{(kf)}$) and the standard entropy change of activation ($\Delta S^{\neq}_{(kf)}$) were determined from the temperature dependence (Figure 8) of the reaction with DMAP, and they were consequently obtained from an Eyring plot (Figure 9). Individual second-order rate constants are listed in Table 6.

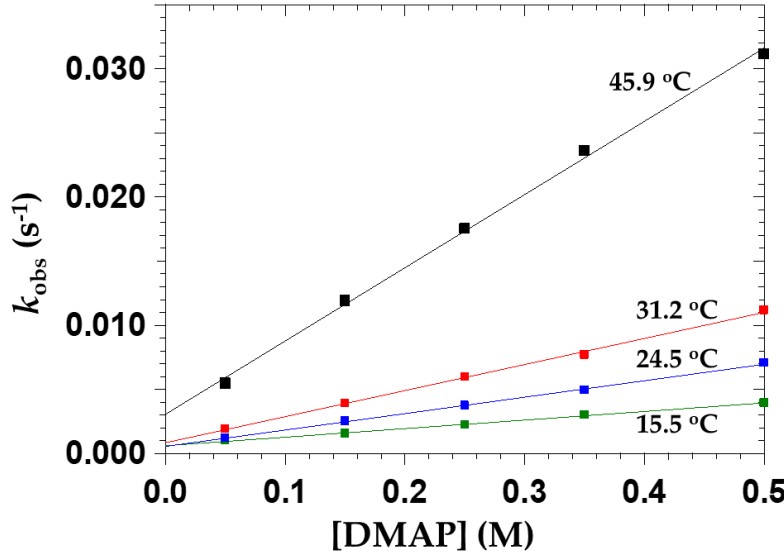

**Figure 8.** Plot of $k_{obs}$ versus [DMAP] for the substitution of triphenylphosphine oxide from *cis*-[NbO(ca)$_2$(H$_2$O)OPPh$_3$]$^-$ by DMAP in acetonitrile at four different temperatures, [Nb] = 0.001 M, $\lambda_{max}$ = 430 nm.

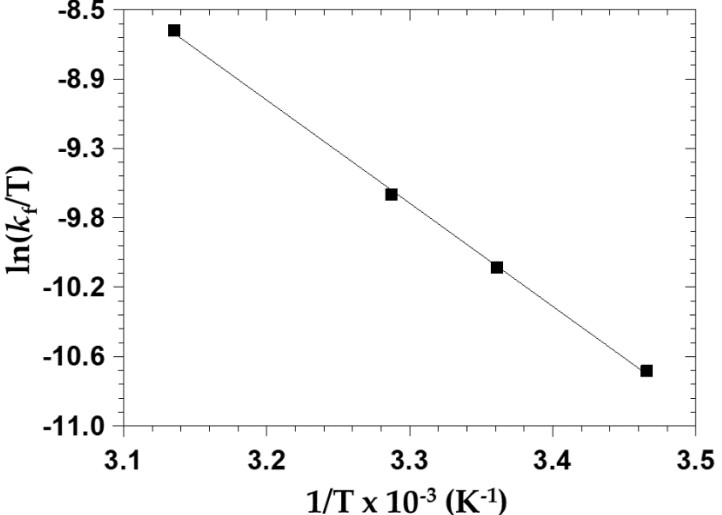

**Figure 9.** Eyring plot for the forward rate constant of the substitution of triphenylphosphine oxide from *cis*-[NbO(ca)$_2$(H$_2$O)OPPh$_3$]$^-$ by DMAP in acetonitrile.

**Table 6.** Summary of the kinetic data for the substitution of triphenylphosphine oxide from *cis*-[NbO(ca)$_2$(H$_2$O)OPPh$_3$]$^-$ by DMAP in acetonitrile at four different temperatures, $\lambda_{max}$ = 430 nm.

|  | 15.5 °C | 24.5 °C | 31.2 °C | 45.9 °C |
|---|---|---|---|---|
| $k_f$ (M$^{-1}$s$^{-1}$) × 10$^{-3}$ | 6.7 ± 0.2 | 12.8 ± 0.3 | 20.3 ± 0.5 | 57.1 ± 1.5 |
| $k_r$ (s$^{-1}$) × 10$^{-3}$ | 0.59 ± 0.06 | 0.54 ± 0.08 | 0.83 ± 0.16 | 3.1 ± 0.5 |
| $K_1$ (M$^{-1}$) $^{(a)}$ | 11.3 ± 1.2 | 24 ± 3 | 24 ± 5 | 19 ± 3 |
| $\Delta H^{\neq}{}_{(kf)}$ (kJ mol$^{-1}$) | - | - | 51.5 ± 1.0 | - |
| $\Delta S^{\neq}{}_{(kf)}$ (J K$^{-1}$ mol$^{-1}$) | - | - | −108 ± 3 | - |

$^{(a)}$ Equation (4).

## 3. Discussion

### 3.1. Synthesis

The complexes were successfully synthesized and isolated in high yields (>80%) as described in Section 4, and all complexes could be well characterized by IR, UV/Vis and ($^1$H-, $^{13}$C{$^1$H}- and $^{31}$P{$^1$H}) NMR spectroscopy as well as X-ray diffraction analysis. However, a significant drawback is considered to be the non-isolation of the final product as defined in Equation (1). Nevertheless, as indicated above, it was characterized spectroscopically and in agreement with the clean kinetics observed. The self-assembly and corresponding isolation of the novel structure of **6**, (Et$_4$N)$_4$[Nb$_4$O$_4$(ca)$_2$($\mu^2$-O)$_2$Cl$_8$]·2CH$_3$CN is specifically significant.

### 3.2. X-ray Crystallography
#### 3.2.1. Correlation of Geometrical Parameters in **2**, **5** and **6**

A number of interesting geometrical aspects emerge when considering the bond distances and angles, as listed in Tables 2–4, and they are summarized next. In general, the rigidity of the ca$^{2-}$ ligand system is prominent and remains fairly unchanged in all three compounds.

The C1-O1 and C2-O2 bond lengths in caH$_2$ (Figure 1, **2a**) are 1.2256 (15) Å and 1.3173 (14) Å, respectively, which corresponds to C=O and C-OH (Table 2), while the C3-Cl1 bond length is 1.7186 (12) Å. The asymmetric unit of (**2**) contains half of a molecule of caH$_2$ that is located on an inversion center, and its symmetry-generated counterparts are linked by hydrogen bonds to the solvated water molecule. This generates an infinite network of hydrogen bonds which can be characterized as strong O-H···O hydrogen bonds, forming a zigzag chain. Inter- and intramolecular hydrogen bonds are observed between O2-H···O8 (1.870 Å) and O8-H3···O1ii (2.33 (4) Å). Sandwich-like π–π stacking is observed between adjacent molecules with a distance of 5.1352 (2) Å (1-x, -y, 1-z) between semi-parallel chloranilic acid ring planes.

As indicated earlier, the niobium(V) center in complex **5a** is coordinated by seven oxygen atoms of two chloranilate dianionic (ca$^{2-}$) bidentate ligands, and one each of a OPPh$_3$ monodentate ligand, an aqua ligand and an oxido group (Figure 1 (**5a**)); see also Tables 3 and 4. It thus represents a seven-coordinate complex with a distorted pentagonal bipyramidal geometry; see Figure 10a [36]. On the other hand, complex **6a** contains four niobium(V) metal centers separated by two bridging oxido ligands, two bridging chloranilato bidentate ligands, eight terminal chlorido ligands and four terminal oxido ligands (Figure 1 (**6a**)). Contrary to **5a**, in **6a**, each niobium atom is six-coordinated to one bridging oxido, two chlorido ligands and one $\mu^2$-chloranilato ligand coordinated to *all four* oxygen atoms of one ca$^{2-}$ bidentate ligand and two terminal oxido groups. The geometry of the Nb(V) metal center is distorted octahedral; see Figure 10b.

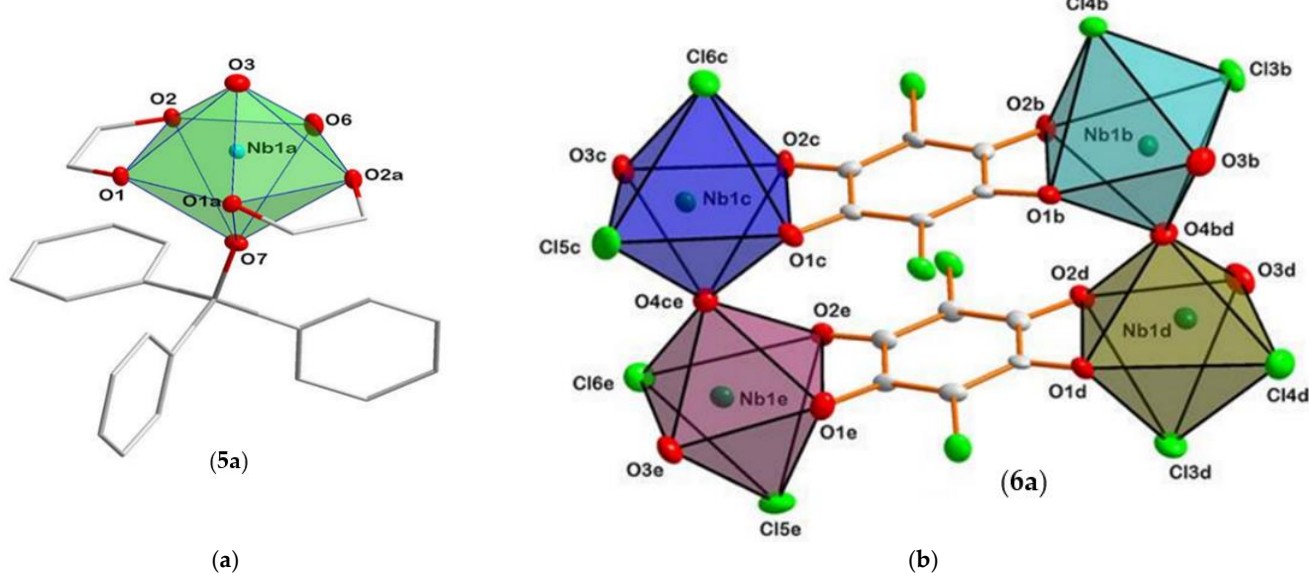

**Figure 10.** Representation of (**a**) the distorted $D_{5h}$ pentagonal bipyramidal coordination polyhedron formed by oxygen donor atoms surrounding the Nb(V) metal centers in *cis*-[NbO(ca)$_2$(H$_2$O)OPPh$_3$]$^-$ (**5a**) and (**b**) a regular octahedral coordination polyhedron surrounding the Nb(V) atoms of [Nb$_4$O$_4$(ca)$_2$($\mu^2$-O)$_2$Cl$_8$]$^{4-}$ (**6a**). The counter ions, solvates and H-atoms are omitted for clarity.

The ca$^{2-}$ ligands in complexes **5a** and **6a** each form a five-membered ring with the average O-Nb-O bite angle being 71.90 (11)° and 70.3 (3)°, respectively, when compared to that of the six-membered metallocycle acetylacetonato complex as previously reported, with an average O-Nb-O bite angle of 80.74 (6)° [32,37,38].

The average Nb-O bond lengths of the $\mu$-chloranilato bidentate ligands in complex **6a** is 2.264 (7) Å, which is longer than the average Nb-O bond length of the chloranilato bidentate ligands in complex **5a** with an average of 2.128 (3) Å. It is worth noting that the one Nb-O bond in **6a** is comparable with the Nb-O bonds in **5a** (ca. 2.11 to 2.14 Å), whereas the other ones are significantly lengthened (ca. 2.39 Å), thus creating a significantly twisted ca$^{2-}$ ligand geometry. In addition, the average Nb-O ca$^{2-}$ ligand bond lengths in complexes **5a** and **6a**, 2.128 (3) Å and 2.264 (7) Å, respectively, are longer than the Nb-O bond distances of the corresponding acetylacetonato complex, where an average of only 1.959 (2) Å was observed [32]. However, a previously reported cupferrate complex with a five-membered metallocycle ring displayed the smallest bite angles, with the O-Nb-O average bite angle of only 69.49 (4)° [2,3]. The average Nb-O bond length of 2.135 (6) Å in the cupferrate complex is slightly longer than the Nb-O distances of complex (**5**) with an average of 2.128 (3) Å and shorter than the Nb-O distances of complex **6** with average of 2.264 (7) Å. The Nb-O(P)$_{trans}$ distance of 2.192 (3) Å in complex **5a** underlines the strong *trans* influence of the *trans* oxido ligand and parallels the spectroscopic evidence.

The average Nb=O bond length in complex **6a** is 1.716 (8) Å and virtually identical to the corresponding bond length in complex **5a** (1.718 (3) Å). The Nb-Nb bond contacts of complex **6a** vary from 3.807 (6) Å to 9.265 (2) Å, while the average Nb-O-Nb distance of 3.814 (2) Å is shorter than the average Nb-ca$^{2-}$-Nb distance of 8.390 (2) Å.

The average C-Cl bond length on the periphery of the ca$^{2-}$ bidentate ligands in complex **5a** is 1.759 (3) Å and slightly shorter than the C-Cl bond length of the free caH$_2$ ligand with the average of 1.7186 (12) Å.

The average bond length of C4a-O8, C5-O11, C5a-O9, and C4-O10 is 1.259 (5) Å, which corresponds to C=O of the uncoordinated site of the ca$^{2-}$ ligand in complex **5a**, while the average bond length of C1a-O1a, C2-O2, C2a-O2a and C1-O1 is 1.299 (4) Å, corresponding to C-O-Nb. However, the average C-O bond lengths of the free ligand (**2**) are 1.2256 (15) Å and 1.3173 (14) Å, which correspond to C=O and C-OH, respectively. When the average

C=O bond lengths of the free ligand are compared to the average bond lengths of C=O and C–O–Nb in the complex **5a**, the C=O average bond length of 1.259 (5) Å is nearly the same as in the free ligand, while the <u>C=O</u>–Nb average bond lengths are slightly shorter than in the free ligand. In the tetranuclear complex **6a**, the average C–Cl bond length of the $\mu$-chloranilato bidentate ligands is 1.728 (10) Å, whereas the average C–Cl bond length of the free caH$_2$ ligand is slightly shorter at 1.7186 (12) Å. Thus, the average C–Cl bond lengths of complex **5a** and complex **6a** are virtually the same.

Finally, the general behavior of the caH$_2$ ligand system is illustrated by the different bond angles, as reported in Table 2. When carefully considering the distortions in the coordinated bidentate ligand (ca$^{2-}$), in both **5a** and **6a**, it follows that the angles indicated by O-(C-C)$_{MC}$ (associated with the Nb-metallocycle) are only ca. 115°, while those on the 'outside' of the ligand, indicated by O-C-**C**$_{Out}$ are ca. 125°, thus about 10° larger, which confirms a significant distortion. The other bond angles in the coordinated bidentate ligand, i.e., (a) those incorporating the chloro substituents on the ring (C-C-**Cl**$_{Out}$) (b) as well as the other C-C-C bond angles 'inside' the ligand hexagonal ring are all virtually identical to the expected 120°.

### 3.2.2. Coordination Geometries

The solid-state structure of **5a** exhibits $\pi$–$\pi$ intramolecular interactions (centroid-centroid distance = 3.552 (1) and 3.788 (2) Å, $-1 + x$, y, z) between the phenyl ring on the OPPh$_3$ and the chloranilate (ca$^{2-}$) ligands, and different triangle and v-shaped bifurcated strong hydrogen bonds were observed in complex **5a**; O-H$\cdots$O, C-H$\cdots$O, O-H$\cdots$Cl, and C-H$\cdots$Cl distances in the range of 1.69 (3)–2.47 (4) Å, 2.40–2.57 Å, 2.52 (4) Å and ca. 2.78 Å, respectively. On the other hand, complex **6a** is stabilized by a tetraethylammonium counter-ion, two acetonitrile solvates and numerous intra- and intermolecular halogen and hydrogen bonding stabilizing interactions. Several different strong halogen (Cl) and hydrogen bonds are observed, with C-H$\cdots$O and C-H$\cdots$Cl distances in the range of 2.42 to 2.60 Å and 2.35 to 2.81 Å, respectively. Three different $\pi$–$\pi$ intramolecular interactions or centroid–centroid distances (3.556 (5) Å, 3.506 (5) Å and 3.576 (5) Å) were observed between the five-membered rings of the $\mu$-chloranilate ligand and the quinone rings in complex **6a**, which also seemingly played important roles in controlling the molecular assembly.

As indicated above, in complex **5a**, the niobium atom is seven coordinated, whereas in complex **6a**, it is six coordinated (Figures 1 and 10) [36]. The distorted $D_{5h}$ pentagonal bipyramidal coordination mode of *cis*-[NbO(ca)$_2$(H$_2$O)OPPh$_3$]$^-$ (Figure 10a) is illustrated within the complex, noting the short oxido ligand distance of 1.721 (1) Å compared to the Nb-O7 distance of 2.197 (1) Å and again emphasizes the large *trans* influence of the oxido ligand. However, a regular octahedral coordination mode within the tetranuclear cluster (Et$_4$N)$_4$[Nb$_4$O$_4$(ca)$_2$($\mu^2$-O)$_2$Cl$_8$]·2CH$_3$CN (**6**) (Figure 10 (**6a**)) is clear, due to different $\mu$-chloranilate ligand coordination modes, such as the coordination modes found in complex **5**, which is unusual for niobium(V) complex structures, as well as for tantalum(V) [39], although it is similar to other hard metals [4,40–42].

### 3.3. Solution Behavior of cis-[NbO(ca)$_2$(H$_2$O)OPPh$_3$]$^-$ (5a): Kinetics and Mechanism

3.3.1. General Spectroscopic Observations

Accurate time-resolved UV/Vis experiments (Figure 2) were performed and allowed a complete analysis of the substitution of the triphenylphosphine oxide from the seven-coordinate complex *cis*-[NbO(ca)$_2$(H$_2$O)OPPh$_3$]$^-$ (**5a**). It yielded clean kinetics, which could be well interpreted once the NMR experiments as described in detail confirmed the final constitution of the product, *cis*-[NbO(ca)$_2$(H$_2$O)(DMAP)]$^-$. Due to the inability to isolate the final product for accurate chemical analysis or single-crystal XRD evaluation, the NMR work was essential.

The $^{31}$P{$^1$H} NMR experiments illustrated in Figures 3 and 4 underline the complicating factors in the substitution process. Therein, free OPPh$_3$ is clearly liberated from **5a** upon the addition of excess water (0.017 M). Moreover, after the addition of DMAP to

*cis*-[NbO(ca)$_2$(H$_2$O)OPPh$_3$]$^-$ (**5a**), the OPPh$_3$ is again (and virtually completely) replaced. However, it is also clear that the *cis* aqua ligand in the reactant *cis*-[NbO(ca)$_2$(H$_2$O)OPPh$_3$]$^-$ species is not substituted by DMAP, only the OPPh$_3$. Thus, from this, it is concluded that the final product is *cis*-[NbO(ca)$_2$(H$_2$O)(DMAP)]$^-$.

The $^1$H NMR spectra presented in Figure 6 confirmed the concentration dependence of DMAP as the entering ligand on the reaction and the changes that are observed in the OPPh$_3$ substitution from *cis*-[NbO(ca)$_2$(H$_2$O)OPPh$_3$]$^-$. It shows that OPPh$_3$ is almost quantitively substituted by DMAP at higher concentrations with an indication of some fast exchange (line broadening) and the dynamic nature of the process. It also confirmed the equilibrium, as indicated in Equation (1).

On the other hand, the tendency of line broadening, when investigated by a preliminary temperature study of the spectral behavior, illustrates the $^1$H NMR spectra. As presented in Figure 7, the *cis*-[NbO(ca)$_2$(H$_2$O)OPPh$_3$]$^-$ complex and equilibria are not significantly affected by temperature. In fact, the shape of the signals is all very similar and indicates either (i) a very fast exchange or (ii) no exchange at all, and that the slight signal shapes observed might be due to proton exchange and not substitution processes.

### 3.3.2. Mechanism of the Substitution of OPPh$_3$ from *cis*-[NbO(ca)$_2$(H$_2$O)OPPh$_3$]$^-$ by a Range of Py-Type Ligands

The systematic study of the substitution of OPPh$_3$ from the starting complex, *cis*-[NbO(ca)$_2$(H$_2$O)OPPh$_3$]$^-$, by five different entering Py-type ligands was conducted wherein a progressive increase in [Py] was studied. The abbreviated stoichiometric mechanism as presented in Equation (1) is supported by the results [24,41].

A comparison of the equilibrium constants $K_1$, p$K_a$ values of the entering nucleophiles, and the rate constants (forward and reverse; Equation (1)) for the substitution of OPPh$_3$ from the starting complex, *cis*-[NbO(ca)$_2$(H$_2$O)OPPh$_3$]$^-$ (**5a**), is illustrated in Table 5 and Figures 7 and 8.

As expected for an *associative* process in hard metal centers, DMAP and 3Brpy (p$K_a$ values of 9.6 and 2.84, respectively) as entering nucleophiles display the fastest and slowest reaction rates of the five Py-type ligands, since they are more electron donating and withdrawing than the 3Clpy, py and 4Mepy. The highest $k_f$ value is observed for DMAP, while a significantly lower value is obtained for 3Brpy, which is in broad agreement with the Brønsted basicity and the electron-donating ability of the two ligands. The following surprising tendency was observed: 3Brpy ≈ 3Clpy ≈ 4Mepy ≈ py < DMAP as defined by $k_f$ = (0.147 ± 0.004) × 10$^{-2}$, (0.206 ± 0.007) × 10$^{-2}$, (0.181 ± 0.006) × 10$^{-2}$, (0.131 ± 0.002) × 10$^{-2}$ and (2.032 ± 0.002) × 10$^{-2}$ M$^{-1}$s$^{-1}$, respectively (Figure 7).

Thus, although the difference between DMAP and 3Brpy is clear, it is interesting to note that a slightly *higher* $k_f$ value is observed for 3Clpy compared to 3Brpy, even though they have similar p$K_a$ values (Figure 7 and Table 5). More surprisingly though, since both py and Mepy have intermediate and higher p$K_a$ values, their second-order rate constants were found to be virtually *identical* to that of the 3Brpy and the 3Clpy. The exact reason for this is not clear with the current data in hand.

However, a more direct trend based on p$K_a$ values is observed when comparing the *first-order* rate constants (Equation (2)) at, e.g., (Py) = 0.35 M, wherein it was found that 3Brpy < 3Clpy < 4Mepy < py < DMAP: 1.19 ± 0.01) × 10$^{-3}$, 1.44 ± 0.01) × 10$^{-3}$, 1.60 ± 0.01) × 10$^{-3}$, 1.66 ± 0.01) × 10$^{-3}$, and (7.68 ± 0.01) × 10$^{-3}$ s$^{-1}$. Since the *first-order* rate constants are dependent on the sum of both the forward and reverse rate constants, this seemingly clearer tendency is obtained.

It is also noteworthy though that the reverse rate constants, $k_r$, vary by less than 1.5 times, i.e., from only (0.07 ± 0.02) × 10$^{-3}$ to (0.12 ± 0.01) × 10$^{-3}$ s$^{-1}$ for all five Py-type ligands. This suggests a virtual *independence* of the reverse reaction (substitution of the Py-type ligand by the OPPh$_3$) in spite of significant decreasing electron-donating abilities when moving in the series DMAP, 4Mepy, py, 3Clpy and 3Brpy.

Nevertheless, it is again in reasonable agreement with an associative activation, wherein the dissociation of the Py-type ligands is not expected to contribute significantly to the activated state. Thus, the attack of the OPPh$_3$ at the Nb(V) in the reverse reaction is seemingly not significantly affected by the strength of the Nb-Py bond, which may overall point to very weak Nb–Py interactions, which is in agreement with the small observed $K_1$ values.

It is noted that the current study, purely from a variation of the entering nucleophiles, although very important and informative, does not give a clear-cut indication on the type of intimate mechanism.

To further try to better answer the question, the temperature study with the DMAP as an entering nucleophile is considered. As indicated in Table 6 and Figure 9, the activation parameters for the OPPh$_3$ substitution reaction from *cis*-[NbO(ca)$_2$(H$_2$O)OPPh$_3$]$^-$ by DMAP were calculated as (51 ± 1) kJ mol$^{-1}$ and (−108 ± 3) J K$^{-1}$ mol$^{-1}$ for the standard change of enthalpy ($\Delta H^{\neq}$(kf)) and the standard change of entropy ($\Delta S^{\neq}$(kf)), respectively. The negative entropy of activation implies a more ordered transition state, and presumably, this substitution reaction involves an *associative* ligand substitution mechanism [24], which is often encountered for classic 16e species. As indicated earlier, this is typical for hard metal centers such as the Nb(V) in this study. Nevertheless, overall, valuable information regarding the substitution process was obtained.

## 4. Materials and Methods

### 4.1. Materials

All reagents used for the synthesis and characterization were of analytical grade and were purchased from Sigma-Aldrich (Johannesburg, South Africa). Some reagents were used as received without further purification. All organic solvents were purified and dried according to the literature [40,43].

### 4.2. Physical Measurements

Infrared spectra of the complexes were recorded on a Bruker Tensor 27 Standard System spectrophotometer (Billerica, MA, USA) utilizing a He-Ne laser at 632.6 nm at a range of 4000–600 cm$^{-1}$. All samples were analyzed as solid-state species via Attenuated Total Reflection (ATR) mode infrared spectrophotometry, and all data were recorded at room temperature. No solution or KX (where X = I, Cl, Br) solid salt pellets were utilized, since halogen interaction was expected from the solution cell and the KX pellet preparation technique. $^1$H, $^{31}$P{$^1$H} and $^{13}$C{$^1$H} NMR spectra were obtained in deuterated CD$_3$OD, CD$_3$CN, and (CD$_3$)$_2$SO on either a Bruker AVANCE II 600 MHz ($^1$H: 600.28 MHz; $^{31}$P{$^1$H}: 242.99 MHz; $^{13}$C: 150.95 MHz) (probe for $^1$H and $^{13}$C{$^1$H} is 5 mm DUAL $^{13}$C-$^1$H/D with z-gradients, while for $^{31}$P{$^1$H} and $^1$H is 5 mm TBI$^1$H/$^{31}$P/D-BB with Z-gradients), or an AVANCE III 400 MHz ($^1$H: 400.13 MHz; $^{13}$C{$^1$H}: 100.61 MHz; $^{31}$P{$^1$H}: 161.97 MHz) (liquid state 5 mm BBI H-BB-D probe with z-gradients) or a FOURIER 300 MHz ($^1$H: 300.18 MHz; $^{13}$C{$^1$H}: 75.48 MHz) (5 mm $^{13}$C/$^1$H high-resolution NMR probe equipped with Z-gradient coil) nuclear magnetic resonance spectrometer operating at 25 °C. Chemical shifts are reported relative to tetramethylsilane using the CD$_3$CN ($^1$H NMR: 1.94 ppm; $^{13}$C NMR: 1.32 and 118.26 ppm), CD$_3$OD ($^1$H NMR: 3.31 ppm; $^{13}$C{$^1$H} NMR: 49.1 ppm), CD$_3$CN ($^1$H NMR: 1.94 ppm; $^{13}$C{$^1$H} NMR: 1.32 and 118.26 ppm) and (CD$_3$)$_2$SO ($^1$H NMR: 2.50 ppm; $^{13}$C{$^1$H} NMR: 39.52 ppm) peaks. $^{31}$P spectra were collected at 360 scans per spectrum, with chemical shifts reported relative to 85% H$_3$PO$_4$ (0 ppm).

### 4.3. Synthesis

**Recrystallization of (caH$_2$)·2H$_2$O (2):** The ligand (**2a**), 2,5-dichloro-3,6-dihydroxy-2,5-cyclohexadiene-1,4-dione (caH$_2$) (0.0750 g, 0.3591 mmol) was obtained from Sigma-Aldrich, Johannesburg, South Africa, and recrystallized from acetonitrile/methanol (3:2) at room temperature. The solution was allowed to stand for a few days after which dark purple, cuboid crystals, suitable for X-ray diffraction were obtained using a slow evaporation technique.

**Synthesis of (Et$_4$N)[NbCl$_6$] (1):** Tetraethylammonium hexachloridoniobate(V), (Et$_4$N)[NbCl$_6$], was prepared by modifying the known procedures [2,41,44]. A solution of NbCl$_5$ (0.2701 g, 1.0 mmol) was carefully (exothermic reaction) dissolved in acetonitrile (10 mL) and then treated with Et$_4$NCl (0.1657 g, 1.0 mmol). The mixture was refluxed overnight at 90 °C, and the solvent was removed under vacuum, yielding a yellow (Et$_4$N)[NbCl$_6$] solid. IR (ATR): $\nu_{(N-C)}$ = 1171 cm$^{-1}$, UV/Vis: $\lambda_{max}$ = 284 nm, $\varepsilon$ = 2.663 × 10$^3$ M$^{-1}$cm$^{-1}$. $^1$H NMR (300.18 MHz, (CD$_3$)$_2$SO): $\delta_{ppm}$ = 3.18–3.26 (q, 8H), 1.13–1.19 (m, 12H). $^{13}$C{$^1$H} NMR (75.48 MHz, (CD$_3$)$_2$SO): $\delta_{ppm}$ = 7.00, 51.29 (Yield: 0.398 g, 94.3%).

**Synthesis of (Et$_4$N)*cis*-[NbO(ca)$_2$(H$_2$O)OPPh$_3$].2H$_2$O.THF (5):** 2,5-dichloro-3,6-dihydroxy-2,5-cyclohexadiene-1,4-dione (caH$_2$) (0.1680 g, 0.8039 mmol) was dissolved in chloroform (10 mL), and Et$_3$N (0.1627 g, 1.6078 mmol) was added dropwise to the clear solution of caH$_2$ upon which the violet color changed to dark brown as a precipitate formed. This was followed by the addition of (Et$_4$N)[NbCl$_6$] (0.1700 g, 0.4020 mmol) dissolved in acetonitrile (10 mL). After 30 min, OPPh$_3$ (0.112 g, 0.402 mmol) was added to the solution at room temperature. The resulting mixture was stirred overnight, after which the volatile material was removed in vacuo. IR(ATR): $\nu_{(Nb=O)}$ = 923 cm$^{-1}$, $\nu_{co}$ = 1537 (s) cm$^{-1}$, UV/Vis: $\lambda_{max}$ = 530 nm, $\varepsilon$ = 2.008 × 10$^3$ M$^{-1}$cm$^{-1}$, $^1$H NMR (600.28 MHz, CD$_3$CN): $\delta_{ppm}$ = 7.57–7.74 (m, 15H, Ph), 3.20–3.36 (m, 8H, Et$_4$N, CH$_2$), 1.29–1.32 (m, 12H, Et$_4$N, CH$_3$). $^{31}$P{$^1$H} (243 MHz, CD$_3$CN): $\delta_{ppm}$ = 38.99. $^{13}$C{$^1$H} NMR (150.95 MHz, CD$_3$CN): $\delta_{ppm}$ = 173.56 (C=O; C-O, ca), 132.40 (*ipso*-Ph), 131.71, 131.65, 129.24, 128.65, 128.55, 110.90 (C-Cl, ca), 51.81 (CH$_2$, Et$_4$N), 6.15 (CH$_3$, Et$_4$N), (Yield: 0.312 g, 81.7%). Crystals suitable for SCXRD were obtained from a solution of MeOH/THF.

**Synthesis of (Et$_4$N)$_4$[Nb$_4$O$_4$(ca)$_2$($\mu^2$-O)$_2$Cl$_8$].2CH$_3$CN (6):** Et$_3$N (0.0967 g, 0.9564 mmol) was added dropwise to a dichloromethane solution of 2,5-dichloro-3,6-dihydroxy-2,5-cyclohexadiene-1,4-dione (caH$_2$) (0.0999 g, 0.4782 mmol in 10 mL CH$_2$Cl$_2$) at 10 °C. After ca. 0.5 h, a solution of (Et$_4$N)[NbCl$_6$] (0.4036 g, 0.9564 mmol in 10 mL CH$_3$CN) was added dropwise to a black–violet clear solution (**careful:** exothermic reaction); the color changed to chocolate brown as a precipitate formed. The resulting mixture was stirred overnight, and the temperature was slowly increased to 40 °C. The volatile material was removed in vacuo. The resulting dark brown residue was washed with diethyl ether (2 × 10 mL), and the Et$_3$NHCl salt was removed from the resulting product by washing with cold chloroform (2 × 10 mL) and dried in vacuo. IR (ATR): $\nu_{(Nb=O)}$ = 949 cm$^{-1}$, $\nu_{CO}$ = 1548 (s) cm$^{-1}$, UV/Vis: $\lambda_{max}$ = 329 nm, $\varepsilon$ = 1.464 × 10$^4$ M$^{-1}$cm$^{-1}$, $^1$H NMR (300.18 MHz, (CD$_3$)$_2$SO): $\delta_{ppm}$ = 3.02–3.21 (m, 32H, Et$_4$N, CH$_2$), 1.15–1.21 (m, 48H, Et$_4$N, CH$_3$). $^{13}$C{$^1$H} NMR (75.48 MHz, (CD$_3$)$_2$SO): $\delta_{ppm}$ = 172.21 (C-O, ca), 103.33 (C-Cl, ca), 51.87, 45.71 (CH$_2$, Et$_4$N), 8.87, 7.64 (CH$_3$, Et$_4$N), (Yield: 0.201 g, 85.6%). Crystals suitable for SCXRD were obtained from a solution of MeOH/THF.

*4.4. X-ray Data Collection, Reduction, and Refinement*

X-ray intensity data were collected on a Bruker X8 Apex II 4K Kappa CCD area detector diffractometer, equipped with a graphite monochromator and MoK$\alpha$ fine focus sealed tube ($\lambda$ = 0.71073 Å, T = 100 (2) K and 298 (2) K) operated at 2.0 kW (50 kV, 40 mA). The initial unit cell determinations and data collections were completed using the APEX2 [42,45] software package. The collected frames were integrated using a narrow-frame integration algorithm and reduced with the Bruker SAINT-Plus and XPREP software packages [43,46], respectively. Analysis of the data showed no significant decay during the data collection. The data were corrected for absorption effects using the multi-scan technique SADABS [44,47], and the structure was solved by the direct methods package SIR-97 [45,48] and refined using the WinGX [46,49] software incorporating SHELXL [47,50]. The final anisotropic full-matrix least-squares refinement was completed on F$^2$. All H-atoms were positioned on geometrically idealized positions and refined using the riding model with fixed C-H distances for aromatic C-H of 0.93 Å (C-H) [U$_{iso}$ (H) = 1.2 U$_{eq}$], for methyl C-H of 0.96 Å (C-H) [U$_{iso}$ (H) = 1.5 U$_{eq}$], for methylene C-H of 0.97 Å (C-H) [U$_{iso}$ (H) = 1.5 U$_{eq}$], for methine C-H of 0.98 Å (C-H)

[$U_{iso}$ (H) = 1.5 $U_{eq}$]. Non-hydrogen atoms were refined with anisotropic displacement parameters. The graphics were completed using the DIAMOND [48,51] program with 50% probability ellipsoids for all non-hydrogen atoms.

**Structure of (Et$_4$N)*cis*-[NbO(ca)$_2$(H$_2$O)OPPh$_3$].3H$_2$O.THF (5):** The purple–violet residue of (Et$_4$N)*cis*-[NbO(ca)$_2$(H$_2$O)OPPh$_3$] was washed with hexane (2 × 10 mL) and dried in vacuo. The complex was dissolved in THF/methanol (1:1), and the solution was left to stand at 25 °C for a few days, after which red cuboid crystals, suitable for X-ray diffraction, were obtained. All solvents (H$_2$O) had to be DFIX restrained to a target value: the distance between H and O fixed at 0.85 Å and the distance between H and H fixed at 1.35 Å to best fit the experimental electron density in Shelx-97. The crystallographic structural characterization of (Et$_4$N)*cis*-[NbO(ca)$_2$(H$_2$O)OPPh$_3$] is represented in Figure 1 (**5a**).

**Structure of (Et$_4$N)$_4$[Nb$_4$O$_4$(ca)$_2$($\mu^2$-O)$_2$Cl$_8$].2CH$_3$CN (6):** The dark brown residue of (Et$_4$N)4[Nb$_4$O$_4$(ca)$_2$($\mu^2$-O)$_2$Cl$_8$] was dissolved in MeOH/THF (2:3), the extracts were filtered through Celite, and the solution was left to stand at room temperature for a few weeks after which brown rhomboid crystals, suitable for X-ray diffraction, were obtained. The high $R_1$ (11%) and $wR_2$ (25%) values indicate that the crystals damaged very fast as soon as they were removed from their mother liquor, even when preventing contact with the atmosphere (or air) using the oil-drop method. One of the counter ions of tetraethylammonium (C36) and solvated acetonitrile (N5) indicated a slight disorder. Overall, the poor quality of the structural model is related to the degradation of the crystals and disordered solvent area; with the crystals at hand, this is the best model accessible. Moreover, the fact that four Nb atoms make up the structure resulted in a significant number of quite heavy metal centers within the anion, thus resulting in some ripple electronic effects, as indicated by the larger residual electron density peaks in the unit cell. The crystallographic structural characterization of (Et$_4$N)$_4$[Nb$_4$O$_4$(ca)$_2$($\mu^2$-O)$_2$Cl$_8$] is represented in Figure 1 (**6a**).

*4.5. Kinetic Experiments*

UV/Vis spectra were recorded using a Varian Cary 50 Conc UV/Visible spectrophotometer equipped with a Julabo F12-mV temperature cell regulator (Seelbach, Germany; accurate within 0.1 °C) in a 1.000 ± 0.001 cm quartz cuvette cell. All the synthesized complexes were completely dried in vacuo (Buchi Rotavapor R-210 Rotary Evaporator (Flawil, Switzerland) and Heidolph Laborota 4000 Rotary Evaporator (Schwabach, Germany). Concentrations of the solutions and the equivalents added in the reactivity and bonding studies were calculated and are expressed with respect to the amount of niobium ion in solution. $^1$H and $^{31}$P{$^1$H} NMR spectra were obtained in deuterated CD$_3$CN on a Bruker AVANCE II 600 MHz ($^1$H: 600.28 MHz; $^{31}$P: 242.99 MHz) (probe for $^1$H and $^{31}$P{$^1$H} and is 5 mm TBI$^1$H/$^{31}$P/D-BB with Z-gradients), AVANCE III 400 MHz ($^1$H: 400.13 MHz; $^{31}$P{$^1$H}: 161.97 MHz) for liquid state 5 mm BBI H-BB-D probe with z-gradients) and FOURIER 300 MHz ($^1$H: 300.18 MHz is 5 mm $^1$H high-resolution NMR probe equipped with Z-gradient coil) nuclear magnetic resonance spectrometers operating at 25 °C. Chemical shifts are reported relative to tetramethylsilane using the CD$_3$CN ($^1$H NMR: 1.94 ppm) peaks. $^{31}$P spectra were collected at 360 scans per spectrum, with chemical shifts reported relative to 85% H$_3$PO$_4$ (0 ppm).

## 5. Conclusions

An important contribution to the coordination chemistry of [Et$_4$N][NbCl$_6$] with oxygen donor ligands such as chloranilic acid is presented, which again underlines the subtle interplay between hard (early) transition metal centers and main group ligand systems. Due to the unstable (e.g., reactive and hydrolysis) nature of the starting material niobium pentachloride, NbCl$_5$, tetraethylammonium chloride, (Et$_4$NCl), was used as a counter ion source to generate the easier-to-handle synthon tetraethylammonium hexachloridoniobate(V), (Et$_4$N)[NbCl$_6$] (**1**). The two novel different niobium(V) compounds, (Et$_4$N)*cis*-

[NbO(ca)$_2$(H$_2$O)OPPh$_3$]-3H$_2$O·THF (**5**; mononuclear, distorted pentagonal bipyramid) and (Et$_4$N)$_4$[Nb$_4$O$_4$(ca)$_2$($\mu^2$-O)$_2$Cl$_8$]·2CH$_3$CN (**6**; tetranuclear) could be successfully synthesized, the latter cluster entity readily obtained by slower self-assembly.

The compounds were successfully characterized by IR, NMR and UV/Vis spectroscopy under atmospheric conditions as well as single crystal X-ray diffraction (SCXRD) at 100 (2) K. These SCXRD determinations of the structures of the complexes, **5a** and **6a**, revealed interesting coordination behavior and π–π stacking interactions as well as an infinite extended network of halogen and hydrogen bonds. Important comparisons were drawn with similar structures from the literature and illustrated the significant chemical and geometric differences between complexes **5a** and **6a**. The [Nb$_4$O$_4$(ca)$_2$($\mu^2$-O)$_2$Cl$_8$]$^{4-}$ complex (**6a**) was substitution inert. However, the *cis*-[NbO(ca)$_2$(H$_2$O)OPPh$_3$]$^-$ (**5a**) complex exhibited clean kinetics and allowed a complete mechanistic investigation of the triphenylphosphine oxide (OPPh$_3$) substitution by a range of pyridine-type ligands. Despite some discrepancies in the kinetic results, in general, it points to an associative activation operative for the substitution process.

This study enabled a better understanding of the physical and chemical behavior of niobium and tantalum metals and their structure/reactivity relationships, and it may prompt further investigation into the separation of the two metals, Nb(V) and Ta(V), and for future application as, e.g., catalysts.

**Supplementary Materials:** The following supporting information can be downloaded at: https://www.mdpi.com/article/10.3390/inorganics10100166/s1, Table S1. Selected bond lengths within the chloranilic acid/chloroanilate ligand in(caH2)·2H2O (**2**), (Et4N)*cis*-[NbO(ca)2(H2O)OPPh3]·3H2O·THF (**5**) and (Et4N)4[Nb4O4(ca)2($\mu$2-O)2Cl8]·2CH3CN (**6**); Table S2. Atomic coordinates (×104) and equivalent isotropic displacement parameters (Å$_2$ × 103) for $\frac{1}{2}$(caH2)·H2O (**2**); Table S3. Bond lengths (Å) and angles (°) for $\frac{1}{2}$(caH2)·H2O (**2**); Table S4. Anisotropic displacement parameters (Å2 × 103) for $\frac{1}{2}$(caH2)·H2O (**2**); Table S5. Hydrogen coordinates (×104), isotropic displacement parameters (Å$_2$ × 103) and general hydrogen-bond distances (Å) and angles (°) of $\frac{1}{2}$(caH2)·H2O (**2**); Table S6. Atomic coordinates (×104) and equivalent isotropic displacement parameters (Å$_2$ × 103) for (Et4N)*cis*-[NbO(ca)2(H2O)OPPh3]·3H2O·THF (**5**); Table S7. Bond lengths (Å) and angles (°) for (Et4N)*cis*-[NbO(ca)2(H2O)OPPh3]·3H2O·THF (5); Table S8. Anisotropic displacement parameters (Å2 × 103) for (Et4N)*cis*-[NbO(ca)2(H2O)OPPh3]·3H2O·THF (5);Table S9. Hydrogen coordinates (×104) and isotropic displacement parameters (Å$_2$ × 103) for (Et4N)*cis*-[NbO(ca)2(H2O)OPPh3]·3H2O·THF (**5**); Table S10. Torsion angles (°) for (Et4N)*cis*-[NbO(ca)2(H2O)OPPh3]·3H2O·THF (**5**); Table S11. General hydrogen-bond distances (Å) and angles (°) of (Et4N)*cis*-[NbO(ca)2(H2O)OPPh3]·3H2O·THF (5); Table S12. Atomic coordinates (×104) and equivalent isotropic displacement parameters (Å$_2$ × 103) for (Et4N)4[Nb4O4(ca)2($\mu$2-O)2Cl8].2CH3CN (**6**); Table S13. Bond lengths (Å) and angles (°) for (Et4N)4[Nb4O4(ca)2($\mu$2-O)2Cl8]·2CH3CN (**6**); Table S14. Anisotropic displacement parameters (Å2 × 103) for (Et4N)4[Nb4O4(ca)2($\mu$2-O)2Cl8]·2CH3CN (**6**); Table S15. Hydrogen coordinates (×104) and isotropic displacement parameters (Å$_2$ × 103) for (Et4N)4[Nb4O4(ca)2($\mu$2-O)2Cl8]·2CH3CN (**6**); Table S16. Torsion angles (°) for (Et4N)4[Nb4O4(ca)2($\mu$2-O)2Cl8]·2CH3CN (**6**); Table S17. General hydrogen-bond distances and angles for (Et4N)4[Nb4O4(ca)2($\mu$2-O)2Cl8]·2CH3CN (**6**); Table S18. Temperature and [DMAP] dependence of the pseudo first-order reaction between *cis*-[NbO(ca)2(H2O)OPPh3]$^-$ (**5a**)and 4-(dimetylamino)pyridine. [Nb] = 1.0 × 10$^{-3}$ M, λ = 430 nm, MeCN; Table S19. Substituent dependence of the pseudo first-order reaction between *cis*-[NbO(ca)2(H2O)OPPh3]$^-$ (**5a**) and pyridine derivative ligands (Py). [Nb] = 1.0 × 10$^{-3}$ M, λ = 430 nm, 31.2 °C, MeCN; Figure S1. ATR Infrared spectrum of caH2.H2O (**2**); Figure S2. ATR Infrared spectrum of (Et4N)[NbCl6] (**1**); Figure S3. ATR Infrared spectrum of (Et4N)*cis*-[NbO(ca)2(H2O)OPPh3]·3H2O·THF (**5**); Figure S4. ATR Infrared spectrum of (Et4N)4[Nb4O4(ca)2($\mu$2-O)2Cl8].2CH3CN (**6**); Figure S5. H-1 NMR spectrum of caH2.H2O (**2**) in DMSO-d6; Figure S6. H-1 NMR spectrum of triphenylphosphine oxide in DMSO-d6; Figure S7. H-1 NMR spectrum of (Et4N)*cis*-[NbO(ca)2(H2O)OPPh3]·3H2O·THF (**5**) in DMSO-d6; Figure S8. H-1 NMR spectrum (Et4N)4[Nb4O4(ca)2($\mu$2-O)2Cl8].2CH3CN (**6**) in DMSO-d6; Figure S9. C-13 NMR spectrum of (Et4N)[NbCl6] (**1**) in DMSO-d6; Figure S10. C-13 NMR spectrum of (Et4N)*cis*-[NbO(ca)2(H2O)OPPh3]·3H2O·THF (**5**) in DMSO-d6; Figure S11. C-13 NMR spectrum of (Et4N)4[Nb4O4(ca)2($\mu$2-O)2Cl8].2CH3CN (**6**) in DMSO-d6; Figure S12. DIAMOND10 view of $\frac{1}{2}$(caH2)·H2O (**2**) with atom numbering system shown and thermal ellipsoids;

Figure S13. Representation of the plane sin $\frac{1}{2}$(caH2)·H2O (**2**); Figure S14. Inter- and intramolecular interactions of $\frac{1}{2}$(caH2)·H2O (**2**); Figure S15. Unit cell for $\frac{1}{2}$(caH2)·H2O (**2**) showing π–π stacking interactions along the c-axis; Figure S16. Molecular structure of *cis*-[NbO(ca)2(H2O)OPPh3] (**5a**) with atom numbering system shown and the thermal ellipsoids drawn at a 50% probability level; Figure S17. Representation of planes through *cis*-[NbO(ca)2(H2O)OPPh3] (**5a**); Figure S18. Representation of *cis*-[NbO(ca)2(H2O)OPPh3] (**5a**) with interplane distances; Figure S19. Packing diagram of (Et4N)*cis*-[NbO(ca)2(H2O)OPPh3].3H2O.THF (**5**) showing two molecular formula per unit cell along b-axis; Figure S20. Expanded packing diagram of *cis*-[NbO(ca)2(H2O)OPPh3] (**5a**) showing extended wing-like orientation chain head to tail π–π stacking interactions along the b-axis; Figure S21. Extended structure of *cis*-[NbO(ca)2(H2O)OPPh3] (**5a**) showing head to tail packing interaction along the b-axis; Figure S22. Extended structure of (Et4N)*cis*-[NbO(ca)2(H2O)OPPh3].3H2O.THF (**5**) showing the intra- and intermolecular interactions; Figure S23. Extended structure of (Et4N)*cis*-[NbO(ca)2(H2O)OPPh3].3H2O.THF (**5**) showing blue dashed lines, indicating different triangle and v-shaped bifurcated hydrogen-bonds found between its symmetry-generated molecules; Figure S24. Tetranuclear structure of the [Nb4O4(ca)2($\mu2$-O)2Cl8] anion of (**6a**); Figure S25. Tetranuclear structure of (Et4N)4[Nb4O4(ca)2($\mu2$-O)2Cl8].2CH3CN (**6**); Figure S26. Representation of the planes through [Nb4O4(ca)2($\mu2$-O)2Cl8] (**6a**); Figure S27. Tetranuclear structure of [Nb4O4(ca)2($\mu2$-O)2Cl8] (**6a**) anion of with the thick blue dashed lines indicating π–π stacking interactions between the $\mu$-chloranilate rings; Figure S28. Tetranuclear structure of [Nb4O4(ca)2(μ2-O)2Cl8] (6a) with space filling; Figure S29. Tetranuclear structure of [Nb4O4(ca)2($\mu2$-O)2Cl8] of (**6a**) showing sheet-like crystal packing along the b-axis. H-atoms, counter ions and solvated molecules are omitted for clarity; Figure S30. Tetranuclear structure of (Et4N)4[Nb4O4(ca)2($\mu2$-O)2Cl8].2CH3CN (**6**) showing the intra- and intermolecular interactions.

**Author Contributions:** Conceptualization, A.R. and J.A.V.; methodology, A.R. and J.A.V.; software, A.R. and O.T.A.; validation, A.N.B., A.R., J.A.V. and O.T.A.; formal analysis, A.N.B.; investigation, A.N.B.; resources, A.R. and J.A.V.; data curation, A.N.B.; writing—original draft preparation, A.N.B.; writing—review and editing, A.N.B., A.R., J.A.V. and O.T.A.; visualization, A.N.B., A.R. and O.T.A.; supervision, J.A.V. and A.R.; project administration, J.A.V. and A.R.; funding acquisition, J.A.V. and A.R. All authors have read and agreed to the published version of the manuscript.

**Funding:** We acknowledge the University of the Free State, Department of Chemistry, the South African National Research Foundation (SA NRF) and the World Academy of Science (TWAS) (UIDs 99782) for financial support. This includes funding under the Swiss-South Africa joint research program (SSAJRP) from the SA NRF (AR: UID: 107802), as well as from the Competitive Program for Rated Researchers of the SA NRF (AR: UID 111698).

**Data Availability Statement:** The crystallographic data sets for compounds **2**, **5** and **6** are available as Supplementary Material and from the Cambridge Structural Data Center with the following codes, CCDC 2201873 (**2**), 2201870 (**5**) and 2201874 (**6**), respectively. Other data and spectroscopy are available as Supplementary Materials, (Tables S1–S19 and Figures S1–S29).

**Acknowledgments:** Sincere thanks are also due to Linette Twigge for the NMR data analysis. Additional acknowledgement further goes to the crystallographic data collection team at the University of the Free State in Bloemfontein, South Africa.

**Conflicts of Interest:** The authors declare no conflict of interest.

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
