# Peer review of "Synthesis, Single Crystal X-ray Structure, Spectroscopy and Substitution Behavior of Niobium(V) Complexes Activated by Chloranilate as Bidentate Ligand"

_inorganics, doi:10.3390/inorganics10100166_

Round 1
Reviewer 1 Report
1) Authors need to decide whether to use UK or US English in manuscript and not a mix of English.
2) Abstract: Could be improved to highlight research topic, objective & conclusion of this research work.
3) Introduction: This chapter could be improved to enhance the research topic. The focus objective of this research work should be expanded and clearly stated as it is not clear.
4) Arrangement for Section 2: Results and Section 3: Discussion needs vast improvement. Section 3: Discussion is too brief and data are not well-discussed.
5) There should be a consistency on the usage of abbreviation. It is not advisable to alternate between abbreviation and its full name throughout the manuscript. Eg: Pyridine(py) is already stated on Pg1; but then it is back to pyridine in Table 5.
6) It is a difficult read in Section 2.mainly due to its arrangement and labelling of sub-sections. Authors should check and have a good look at the PDF copy of the manuscript (pg 3-16). A careful formatting and proper arrangement is urgently needed here.
7) (A)Pg 9, line 172: It is mentioned in text … entering nucleophiles/ligands (see below)
(B) Pg9, line 178:. Experiments are described below:
What does this ‘see below’ or ‘below’ refer to? Authors need to state and mention clearly. There are several more of this ‘unclear’ statements in section 2: Results.
8) Pg 11, line 265: What is G in Figure 4 as mentioned here?
9) Most figures size, resolution need to be improved. The font size is also inconsistent. There are several elongated figures. Eg Figure 3, 4, 5, 6, 7, 8, 9, 10
10) Pg7, Table 2: A nearly repeated data in table A1 is included in Supplementary materials. This is rather redundant.
11) Crystal structure for compound 6 is not satisfactory and badly disordered. Authors should consider to not to include the data in this manuscript.
12) Figure 1 show that compound 5 is disordered in the phenyl rings. This figure might need improvement and explained.
13) Both structures in figure 10 needs to be of similar size.
14) Conclusion need to be improved to show the significance and relevance of the research work.
15) Reference list could perhaps be expanded and updated to include more recent/comprehensive articles in niobium and tantalum studies.

Author Response
We thank the reviewer for many constructive and critical comments, and modified the ms according to instructions and hope it is now acceptable.
1) Authors need to decide whether to use UK or US English in manuscript and not a mix of English.
- Corrected as per US English
2) Abstract: Could be improved to highlight research topic, objective & conclusion of this research work.
- Modified as indicated
3) Introduction: This chapter could be improved to enhance the research topic. The focus objective of this research work should be expanded and clearly stated as it is not clear.
- Modified as indicated
4) Arrangement for Section 2: Results and Section 3: Discussion needs vast improvement. Section 3: Discussion is too brief and data are not well-discussed.
- Modified as per reviewer comment
5) There should be a consistency on the usage of abbreviation. It is not advisable to alternate between abbreviation and its full name throughout the manuscript. Eg: Pyridine(py) is already stated on Pg1; but then it is back to pyridine in Table 5.
- Corrected; apologies for this oversight.
6) It is a difficult read in Section 2.mainly due to its arrangement and labelling of sub-sections. Authors should check and have a good look at the PDF copy of the manuscript (pg 3-16). A careful formatting and proper arrangement is urgently needed here.
- Modification introduced as indicated.
7) (A)Pg 9, line 172: It is mentioned in text … entering nucleophiles/ligands (see below)
(B) Pg9, line 178:. Experiments are described below: What does this ‘see below’ or ‘below’ refer to? Authors need to state and mention clearly. There are several more of this ‘unclear’ statements in section 2: Results.
- Correct reference to paragraphs and tables included.
8) Pg 11, line 265: What is G in Figure 4 as mentioned here?
- Corrected
9) Most figures size, resolution need to be improved. The font size is also inconsistent. There are several elongated figures. Eg Figure 3, 4, 5, 6, 7, 8, 9, 10
- We corrected the fonts to be of consistent size.
We also introduced new schemes.
Regarding the elongation as per comment: we argued that by artificially increasing the y-direction, additional space in the published version is taken up. The best resolution of the spectra as based on experimental were presented.
10) Pg7, Table 2: A nearly repeated data in table A1 is included in Supplementary materials. This is rather redundant.
- It is in our view that additional explanation from where the average values come from and how used, should be done in the Supplementary information. Complete lists of values are however required on hand to allow the reader to correctly find the origin of the values used.
11) Crystal structure for compound 6 is not satisfactory and badly disordered. Authors should consider to not to include the data in this manuscript.
- We respectfully disagree. The Figure (of the tetrameric anion) as included shows no disorder. There are however disorders in the cations; typical when the Et4N+ cation is used. We nevertheless amended the figure slightly to even better illustrate the geometry of the complex anion.
The structure is in our view a very good one, forms an essential and integral part of the paper and cannot be omitted.
12) Figure 1 show that compound 5 is disordered in the phenyl rings. This figure might need improvement and explained.
- We respectfully do not agree with a formal disorder. Again, it is quite usual that phenyl wings of P(OPh)3 shows significant dynamics and thermal motion. It is considered acceptable as is. A comment has been added as indicated in the Caption.
13) Both structures in figure 10 needs to be of similar size.
- We are unsure exactly what the reviewer wants in this situation. The figures only illustrate the coordination polyhedra and to have an (a) and (b) in the same figure and keep the visual bond distances in perspective is unnecessary in this case. But we decreased the size of (a) slightly as suggested by the reviewer.
14) Conclusion need to be improved to show the significance and relevance of the research work.
- Amended as indicated.
15) Reference list could perhaps be expanded and updated to include more recent/comprehensive articles in niobium and tantalum studies.
- Amended by including additional references as indicated.
Conclusion, there is a number of formatting inconsistencies in the current manuscript. The manuscript needs some careful revision on scientific information and arrangement of data
- Amended as far as possible according to the layout given by INORGANICS
Reviewer 2 Report
The work submitted for consideration in INORGANICS entitled “Synthesis, Single Crystal X-ray Structure, Spectroscopy and Substitution Behaviour of Niobium(V) Complexes Activated by Chloranilate as Bidentate Ligand” by Alebel Nibret Belay, Johan Andries Venter, Orbett Teboho Alexander and Andreas Roodt is a descriptive analysis of Nb(V) catalysts (if future tests are done), using chloroanilates as ligands. The stacked structures with oxo bridges is interesting, and actually those complexes could have properties closer to catalysts anchored on surfaces.
The description of the complexes is good, and maybe I would miss the application of those complexes as catalysts, but this will be a future step forward. Structurally I have the doubt why being Nb(V) the authors find out hexa-coordination around the metal, instead of seven. The authors could comment on this, specifically on niobium(V) DOI: 10.1002/cctc.201200916; 10.1016/j.mcat.2017.10.023) or more general, including niobum(VI) as well (DOI: 10.1002/chem.200390145). And then it would be necessary to point out that the “stacked” structures could go close to the material science (DOI:10.1021/jacs.5b02872), and that the fluxional character of niobium is similar to the tantalum one (DOI: 10.1021/om300437e). However, all this is simply to get more impact and give visibility to this nice crystallographic nice characterization of Nb(V) complexes.
I encourage publication of the paper in Inorganics. And I do not click on “HIGH Significance of content because of the lack of catalysis and the links to catalysis.
Author Response
1) The description of the complexes is good, and maybe I would miss the application of those complexes as catalysts, but this will be a future step forward.
- We thank the reviewer for the kind comment. Indeed, catalysis is potential future research.
- We included the reference and a short sentence to this effect.
2) Structurally I have the doubt why being Nb(V) the authors find out hexa-coordination around the metal, instead of seven. The authors could comment on this, specifically on niobium(V) DOI: 10.1002/cctc.201200916; 10.1016/j.mcat.2017.10.023) or more general, including niobum(VI) as well (DOI: 10.1002/chem.200390145).
- We respectfully disagree. Nb(V) is diamagnetic, as clearly indicated by the NMR data. It is known that the coordination of ligands can induce different geometries.
- References included and a short sentence to this effect as per suggestion; reviewer thanked for the suggestion.
3) And then it would be necessary to point out that the “stacked” structures could go close to the material science (DOI:10.1021/jacs.5b02872), and that the fluxional character of niobium is similar to the tantalum one (DOI: 10.1021/om300437e). However, all this is simply to get more impact and give visibility to this nice crystallographic nice characterization of Nb(V) complexes.
References included and a short sentence to this effect as per suggestion; reviewer thanked for the suggestion.
Round 2
Reviewer 1 Report
1) Authors should really check carefully and compare the values tabulated in Table 2 and Table A1 in Supplementary Materials. The submitted cif files are in good agreement with the tabulated values in the current manuscript while those in Table A1 are still showing the values from 1st manuscript draft. Due to the difference in quoted values, it is recommended to confirm bond lengths and bond angles’ values as described in Section 3.2.1 and 3.2.2.
2) Pg 12. Section 2.3.3.1: it should be 1H NMR studies.
3) Pg 15: Section 2.3.6: ‘cis’ should be in italic.

Author Response
Reviewer 1 |
Author response |
1) Authors should really check carefully and compare the values tabulated in Table 2 and Table A1 in Supplementary Materials. The submitted cif files are in good agreement with the tabulated values in the current manuscript while those in Table A1 are still showing the values from 1st manuscript draft. Due to the difference in quoted values, it is recommended to confirm bond lengths and bond angles’ values as described in Section 3.2.1 and 3.2.2. Conclusion, the manuscript needs some careful checking on the tabulated data. |
We thank the Reviewer for the comments. We have re-checked the complete ms and corrected the Supplementary Materials. In addition, we made a number of smaller corrections to text, punctuation and amendments; not indicated. |
This has been extensively done and carefully checked; a new version of the Supplementary Material has been submitted |
|
2) Pg 12. Section 2.3.3.1: it should be 1H NMR studies. |
Corrected |
3) Pg 15: Section 2.3.6: ‘cis’ should be in italic. |
Corrected throughout ms |
I therefore suggest Minor Revision. |
We thank the Reviewer for the comment |